# Type I IFN exacerbates disease in tuberculosis-susceptible mice by inducing neutrophil-mediated lung inflammation and NETosis

Lúcia Moreira-Teixeira [1✉], Philippa J. Stimpson[1], Evangelos Stavropoulos[1], Sabelo Hadebe [1], Probir Chakravarty[2], Marianna Ioannou [3], Iker Valle Aramburu[3], Eleanor Herbert [4,5], Simon L. Priestnall [4,5], Alejandro Suarez-Bonnet [4,5], Jeremy Sousa[6,7,8], Kaori L. Fonseca[6,7,8,9], Qian Wang[3], Sergo Vashakidze[10], Paula Rodríguez-Martínez[11], Cristina Vilaplana [12], Margarida Saraiva [6,7,14], Venizelos Papayannopoulos[3,14] & Anne O'Garra [1,13,14]

Tuberculosis (TB) is a leading cause of mortality due to infectious disease, but the factors determining disease progression are unclear. Transcriptional signatures associated with type I IFN signalling and neutrophilic inflammation were shown to correlate with disease severity in mouse models of TB. Here we show that similar transcriptional signatures correlate with increased bacterial loads and exacerbate pathology during *Mycobacterium tuberculosis* infection upon GM-CSF blockade. Loss of GM-CSF signalling or genetic susceptibility to TB (C3HeB/FeJ mice) result in type I IFN-induced neutrophil extracellular trap (NET) formation that promotes bacterial growth and promotes disease severity. Consistently, NETs are present in necrotic lung lesions of TB patients responding poorly to antibiotic therapy, supporting the role of NETs in a late stage of TB pathogenesis. Our findings reveal an important cytokine-based innate immune effector network with a central role in determining the outcome of *M. tuberculosis* infection.

[1] Laboratory of Immunoregulation and Infection, The Francis Crick Institute, London NW1 1AT, UK. [2] Bioinformatics and Biostatistics Team, The Francis Crick Institute, London NW1 1AT, UK. [3] Laboratory of Antimicrobial Defence, The Francis Crick Institute, London NW1 1AT, UK. [4] Department of Pathobiology and Population Sciences, Royal Veterinary College, Hatfield AL9 7TA, UK. [5] Experimental Histopathology Team, The Francis Crick Institute, London NW1 1AT, UK. [6] i3S - Instituto de Investigação e Inovação em Saúde, Universidade do Porto, Porto, Portugal. [7] IBMC - Instituto de Biologia Molecular e Celular, Universidade do Porto, Porto, Portugal. [8] ICBAS - Instituto de Ciências Biomédicas Abel Salazar, Universidade do Porto, Porto, Portugal. [9] Programa de Pós-Graduação Ciência para o Desenvolvimento (PGCD), Instituto Gulbenkian de Ciência (IGC), Oeiras, Portugal. [10] National Center for Tuberculosis and Lung Diseases (NCTLD), 50, Maruashvili Str, 0101 Tbilisi, Georgia. [11] Pathology Department, Hospital Universitari Germans Trias i Pujol, Universitat Autònoma de Barcelona (UAB), Crtra. Del Canyet, s/n, 08916 Badalona, Catalonia, Spain. [12] Experimental Tuberculosis Unit (UTE), Fundació Institut Germans Trias i Pujol (IGTP), Universitat Autònoma de Barcelona (UAB), CIBER Enfermedades Respiratorias. Edifici Laboratoris de Recerca. Can Ruti Campus, Crtra. de Can Ruti, Camí de les Escoles, s/n, 08916 Badalona, Catalonia, Spain. [13] National Heart and Lung Institute, Faculty of Medicine, Imperial College London, London W2 1PG, UK. [14]These authors jointly supervised this work: Margarida Saraiva, Venizelos Papayannopoulos, Anne O'Garra. ✉email: luciamoreirateixeira@gmail.com

Tuberculosis (TB) remains a major health problem world-wide with 1.2 million reported deaths in 2018[1]. Despite its clinical significance, there are still significant gaps in our understanding of the mechanisms underlying protection or pathogenesis following *Mycobacterium tuberculosis* infection that could be exploited to improve TB treatment and thus reduce disease severity and mortality[2–4].

Transcriptional studies of human TB have been instrumental in unveiling new mediators of the immune response to *M. tuberculosis* infection[5]. Blood transcriptional signature of active TB revealed a dominance of type I IFN-inducible genes[6] (and reviewed in[7]), which correlated with the extent of radiographic lung disease and was diminished upon successful treatment[6,8,9]. These studies highlighted an important role for type I IFN in driving TB pathogenesis. Consistently, studies in mouse models of TB demonstrated that induction of high and sustained levels of type I IFN by different mechanisms result in increased host susceptibility to *M. tuberculosis* infection[7,10–15]. Transcriptomic analyses in human and mouse studies have also revealed an overabundance of type I IFN and granulocyte-associated genes correlating with lung disease severity in both active TB patients and TB-susceptible mice[16]. In humans, neutrophils are the predominant cell type infected with *M. tuberculosis* in the airways and are detected within inflammatory lung lesions of patients with active disease[17,18]. In mice, neutrophils are abundant in the lung lesions of *M. tuberculosis*-infected susceptible mice and have also been reported to contribute to TB progression promoting both tissue damage and bacterial replication[18–20]. However, the mechanisms of neutrophil-mediated pathology in TB-susceptible mice and patients with active TB remain unclear.

Macrophages and CD4+ T cells, and several cytokines such as IL-12, IFN-γ, TNF and IL-1 have been shown to play a major role in protection against *M. tuberculosis* infection[2–4,13,21–24]. More recently, GM-CSF has also been increasingly recognised as a key mediator for TB resistance[25–31], and is in the pipeline for host-directed therapy as an adjuvant treatment of TB[32]. The presence of GM-CSF specific autoantibodies that block GM-CSF function have been linked to cryptococcal disease and some cases of pulmonary TB[28]. In addition, GM-CSF treatment of mouse and human macrophages restricts intracellular mycobacterial growth in vitro[25,26,29,30] and mice lacking GM-CSF are highly susceptible to *M. tuberculosis* infection[27,31]. Impaired innate immune responses and decreased recruitment of protective IFN-γ producing CD4+ T cells into the site of infection have been proposed as mechanisms for the increased susceptibility following *M. tuberculosis* infection in mice deficient in GM-CSF[27,31]. However, these studies are difficult to interpret owing to the developmental defects in alveolar macrophages and abnormalities in surfactant recycling in mice lacking GM-CSF from birth, leading to the development of a lung disease that resembles human pulmonary alveolar proteinosis (PAP)[33,34]. Therefore, the mechanisms underlying the pathology resulting from an absence of GM-CSF signalling during *M. tuberculosis* infection remain poorly understood.

To further investigate the mechanisms underlying the pathology and disease exacerbation resulting from an absence of GM-CSF signalling, without the confounding developmental lung pathologies resulting from gene deletion, here we instead administer blocking antibodies against GM-CSF during *M. tuberculosis* infection. We show that increased expression of genes associated with neutrophil recruitment and activation, as well as type I IFN-inducible genes, in blood and lungs of *M. tuberculosis*-infected mice in the absence of GM-CSF signalling, accompanies enhanced mycobacterial growth and lung pathology. Disease exacerbation driven by GM-CSF blockade is neutrophil-dependent and associates with exacerbated neutrophil extracellular trap (NET) formation at the site of infection.

Mechanistically, our findings reveal that increased type I IFN signalling induces pulmonary NETosis and promotes mycobacterial growth in the absence of GM-CSF. We show that type I IFN-induced NETosis is also associated with disease severity in TB-susceptible C3HeB/FeJ mice, supporting a more generalised role of type I IFN-induced NETosis in TB pathogenesis over and above GM-CSF blockade. Importantly, we demonstrate for the first time that NETs are present in necrotic lung lesions from patients with nonresolving pulmonary TB, supporting the clinical revelance of NETs in TB pathogenesis.

## Results

**GM-CSF blockade increases host susceptibility to TB.** To address the mechanisms of pathogenesis resulting from *M. tuberculosis* infection in the absence of GM-CSF signalling, without the confounding factors associated with the lack of GM-CSF during host development[33,34], we used monoclonal antibodies (mAbs) directed against GM-CSF (αGM-CSF) to block GM-CSF function during the course of infection. Mice were treated with αGM-CSF or isotype control (labelled as Ctrl Ab) mAbs the day prior to aerosol infection with the *M. tuberculosis* clinical isolate, HN878, and then twice weekly for 3 weeks after infection (Fig. 1a). GM-CSF blockade during *M. tuberculosis* infection resulted in enhanced host susceptibility, similar to the findings reported for GM-CSF-deficient mice[27,31], with significant weight loss (Fig. 1b), increased lung mycobacterial loads (Fig. 1c) and exacerbated lung pathology (Fig. 1d and Supplementary Data 1), when compared to infected Ctrl Ab treated mice. Scoring of histological sections showed an increase in lung lesion burden and severity in infected mice treated with αGM-CSF mAbs (Fig. 1e and Supplementary Data 1). This was characterized by enhanced necrosis and intra-alveolar necrotic debris (Fig. 1f and Supplementary Data 1), with greater numbers of acid-fast bacilli in the lungs of *M. tuberculosis*-infected mice treated with αGM-CSF mAbs as compared to infected Ctrl Ab treated mice (Fig. 1g).

Since increased susceptibility to *M. tuberculosis* infection in GM-CSF-deficient mice has been associated with an impaired T cell-mediated IFN-γ-response at the site of infection[27], we evaluated the effects of αGM-CSF treatment on T cells in the lungs of infected mice. We observed a slight but not significant decrease in the percentage of CD4+ and CD8+ T cells in the lungs of infected mice treated with αGM-CSF compared to Ctrl mAbs, with no effect on T cell numbers (Supplementary Fig. 1a, b). Moreover, similar percentages and numbers of IFN-γ-producing CD4+ and CD8+ T cells were detected ex vivo from both αGM-CSF and Ctrl Ab treated mice (Supplementary Fig. 1a, b), demonstrating that blockade of GM-CSF did not impact IFN-γ responses in infected mice. Blockade of GM-CSF during *M. tuberculosis* infection also resulted in increased lung mycobacterial loads in IFN-γ receptor-deficient (IFNγR−/−) mice (Supplementary Fig. 1c) and in WT mice co-treated with anti-IFN-γ mAbs (Supplementary Fig. 1d), compared to their respective infected Ctrl Ab treated mice. Our findings thus reveal an additional in vivo role for GM-CSF in host resistance to *M. tuberculosis* infection that is independent of IFN-γ.

**GM-CSF controls blood transcriptional TB signatures.** To investigate pathways underlying the pathogenesis resulting upon *M. tuberculosis* infection in the absence of GM-CSF signalling, we performed RNA-Seq of whole blood from uninfected and HN878-infected mice treated with αGM-CSF or Ctrl mAbs. Principal component (PC) analysis depicted distinct global transcriptional signatures in blood samples from the different groups (Supplementary Fig. 2a), with PC1 explaining ~73% of the variance and segregating the uninfected from HN878-infected samples. On the

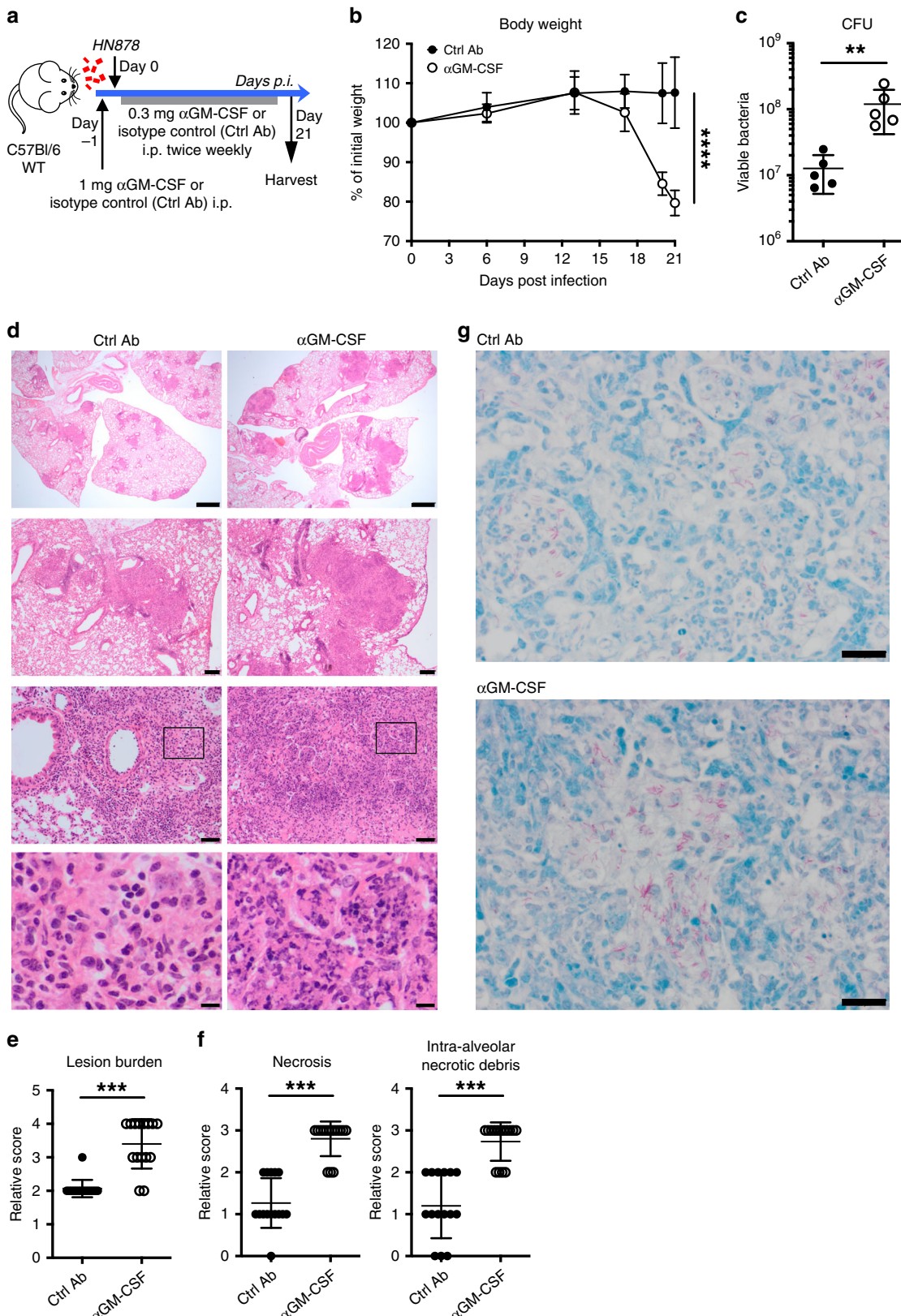

other hand, PC2 segregates the infected samples according to treatment (αGM-CSF versus Ctrl Ab, ~13% of the variance explained) (Supplementary Fig. 2a, b). We identified 7010 genes differently expressed in the blood of HN878-infected Ctrl Ab mice compared to uninfected Ctrl Ab treated mice (Supplementary

Data 2 and Supplementary Fig. 2c, left panel). While we observed negligible transcriptional changes due to αGM-CSF treatment in the blood from uninfected mice (Supplementary Fig. 2c, middle panel), GM-CSF blockade during infection triggered major transcriptional changes in the blood, with 4479 differently expressed

**Fig. 1 GM-CSF controls lung bacterial growth and necrotic inflammation. a** Schematic of experimental design for GM-CSF blockade during *M. tuberculosis* infection: intraperitoneal (i.p.) injection of 1 mg anti-GM-CSF (αGM-CSF; open circles) or isotype control (labelled as Ctrl Ab; closed circles) monoclonal antibodies (mAbs) the day prior to aerosol infection with HN878 strain, followed by i.p. injections of 0.3 mg of αGM-CSF or Ctrl mAbs, twice weekly for 3 weeks. **b** Change in body weight at various days post-infection (*n* = 8 mice/group; mean ± SD). ***$P < 0.0001$. **c** Viable bacterial loads determined in the lungs at day 21 post-infection. **$P = 0.0079$. **d** Representative hematoxylin and eosin (H&E) staining of infected lung sections (*n* = 3 mice/group). Scale bars from top to bottom: 1 mm; 200 μm; 50 μm, 10 μm. **e** Relative lung lesion burden from H&E stained sections (Supplementary Data 1). 0 = no lesions, 1 = focal lesion, 2 = multiple focal lesions, 3 = one or more focal severe lesions, 4 = multiple focal lesions that are extensive and coalesce, 5 = extensive lesions that occupy the majority of the lung lobe. ***$P < 0.0001$. **f** Relative necrosis (left) and intra-alveolar necrotic debris (right) scores from H&E stained lung sections (Supplementary Data 1). 0 = not present, 1 = minimal, 2 = mild, 3 = moderate, 4 = marked. ***$P < 0.0001$. **g** Representative Ziehl–Neelsen staining of infected lung sections (*n* = 3 mice/group). Scale bars, 20 μm. Data representative (**b–d**, **g**) or pooled (**e**, **f**) of five biological experiments. Represented is the mean ± SD; each dot represents an individual mouse: *n* = 5 mice/group (**c**) or *n* = 15 mice/group (**e**, **f**). Source data are provided as a Source Data file. Statistical analysis was performed using two-way ANOVA (**b**) or two-tailed Mann–Whitney test (**c**, **e**, **f**).

genes detected between αGM-CSF versus Ctrl Ab treated infected mice (Supplementary Data 2 and Supplementary Fig. 2c, right panel).

To gain insight into the changes in gene expression caused by GM-CSF blockade during *M. tuberculosis* infection, we tested our previously reported blood modular TB signature, which distinguishes TB patients from healthy controls[35] and is recapitulated in TB-susceptible strains of mice[16], on our blood RNA-Seq data from infected and uninfected αGM-CSF versus Ctrl Ab treated mice (Supplementary Data 3 and Fig. 2a). We detected a weak modular signature in the blood from infected mice (Fig. 2a), qualitatively similar to the human TB signature[35]. Interestingly, the blood modular immune signature observed in αGM-CSF treated infected mice was more enriched than in Ctrl Ab treated infected mice and better resembled the blood signature of active human TB disease (Fig. 2a). Myeloid and granulocyte-associated modules HB3, HB5, HB8, HB14 and HB23 previously described as over-abundant in the blood from TB patients[35] and mouse models of increased susceptibility to TB[16], showed increased over-abundance in the blood from infected mice treated with αGM-CSF (Fig. 2a), shown quantitatively by Eigengene expression (Fig. 2b). This is in keeping with cellular deconvolution analysis[36] showing an increase in the percentage of myeloid and neutrophil fractions in the blood from infected αGM-CSF compared to infected Ctrl Ab treated mice (Supplementary Fig. 3a), which was reflected by increased expression of neutrophil, monocyte and macrophage-associated genes induced by GM-CSF blockade during infection (Supplementary Data 2 and Supplementary Fig. 3b–d), and was similar to our previous report of the blood signature in human TB and TB-susceptible mice[16]. Similarly, the two IFN-associated modules, HB12 and HB23, also previously reported as over-abundant in the blood from patients with active TB disease[35] and from mouse models of increased TB susceptibility[16], were over-abundant in the blood from infected Ctrl Ab treated mice and further increased in the blood from αGM-CSF treated mice (Fig. 2a, b). This is consistent with the presence of IFN-inducible genes among the genes upregulated by GM-CSF blockade during infection (Supplementary Data 2 and Supplementary Fig. 2c, cluster II). In contrast, the cell cycle module (HB13) was strongly over-abundant upon infection but less so in the blood from αGM-CSF treated mice (Fig. 2a). Similarly to the human TB blood signature and that of TB-susceptible mice[16], the T cell and B cell modules (HB2, HB4, HB15 and HB21) were slightly less abundant in the blood from infected mice treated with αGM-CSF mAbs compared to infected Ctrl Ab treated mice (Supplementary Data 3 and Fig. 2a). In keeping with this, cellular deconvolution analysis[36] showed a decrease in the percentage of B cells and CD4$^+$ T cell fractions in the blood from infected αGM-CSF compared to infected Ctrl Ab treated mice (Supplementary Fig. 3a), which was also reflected by decreased expression of key T and B cell-specific genes (Supplementary Data 2 and Supplementary Fig. 2c, cluster V).

**GM-CSF controls lung transcriptional TB signatures**. To determine the transcriptional response at the site of infection, RNA-seq data were obtained from whole lungs of the same mice used for the blood data from Fig. 2a. Sample distribution of lung RNA-Seq data PC analysis highly resembled blood RNA-Seq data, with the biggest transcriptional variance explained by the infection (PC1; Supplementary Fig. 4a, b) and with 11,532 genes differentially expressed between the lung samples from HN878-infected Ctrl Ab treated mice versus the lung samples from uninfected Ctrl Ab treated mice (Supplementary Data 4 and Supplementary Fig. 4c, left panel). Although with a smaller explained variance (~3%), PC2 clearly segregated the infected samples according to αGM-CSF or Ctrl Ab treatment (Supplementary Fig. 4a, b). GM-CSF blockade triggered transcriptional changes in the lungs from both uninfected and infected mice (Supplementary Data 4 and Supplementary Fig. 4c, middle and right panel). In uninfected mice, we detected 3 significantly upregulated and 327 downregulated genes between αGM-CSF versus Ctrl Ab groups (Supplementary Data 4 and Supplementary Fig. 4c, middle panel). Among the genes downregulated by GM-CSF blockade at steady-state, we identified genes associated with alveolar macrophages[37] (Supplementary Data 4 and Supplementary Fig. 4c, middle panel), in keeping with a key role for GM-CSF in the maintenance of these cells[38,39]. During infection, GM-CSF blockade affected the lung expression of 2119 genes, with alveolar macrophage-associated genes among the top down-regulated and neutrophil-related genes among the top upregulated genes (Supplementary Data 4 and Supplementary Fig. 4c, right panel).

To identify GM-CSF driven changes in co-expressed sets of genes across the infected lungs, we next tested the previously published mouse lung disease modular signature[36] on our lung TB RNA-Seq data (Supplementary Data 5 and Fig. 2c). We observed a similar modular signature in the lungs of HN878-infected mice treated with Ctrl Ab (left) or αGM-CSF (right) mAbs, but with distinct pattern in the abundance of certain modules (Fig. 2c). The Type I IFN/*Ifit*/*Oas* module (L5) was over-abundant in the infected lungs but slightly more enriched in infected mice treated with αGM-CSF compared to Ctrl Ab (Fig. 2c, d). This is consistent with an increased expression of IFN-inducible genes resulting from GM-CSF blockade during infection (Supplementary Data 4 and Supplementary Fig. 4c, cluster III). Modules associated with innate inflammation (L10–L13) were also over-abundant in infected lungs but further over-abundance was observed in lungs from infected mice treated with αGM-CSF mAbs compared to infected Ctrl Ab treated mice, also shown quantitatively by Eigengene expression (Fig. 2c, d). The elevated expression of inflammatory genes is consistent with the exacerbated lung pathology observed in HN878-infected αGM-CSF treated mice compared to infected Ctrl Ab treated mice (Fig. 1d–f and Supplementary Data 1). In contrast, the modules *Ifng*/*Gbp*/antigen presentation (L7) and Cytotoxic/T cells/ILC/*Tbx21*/*Eomes*/B cells (L35), which have been associated

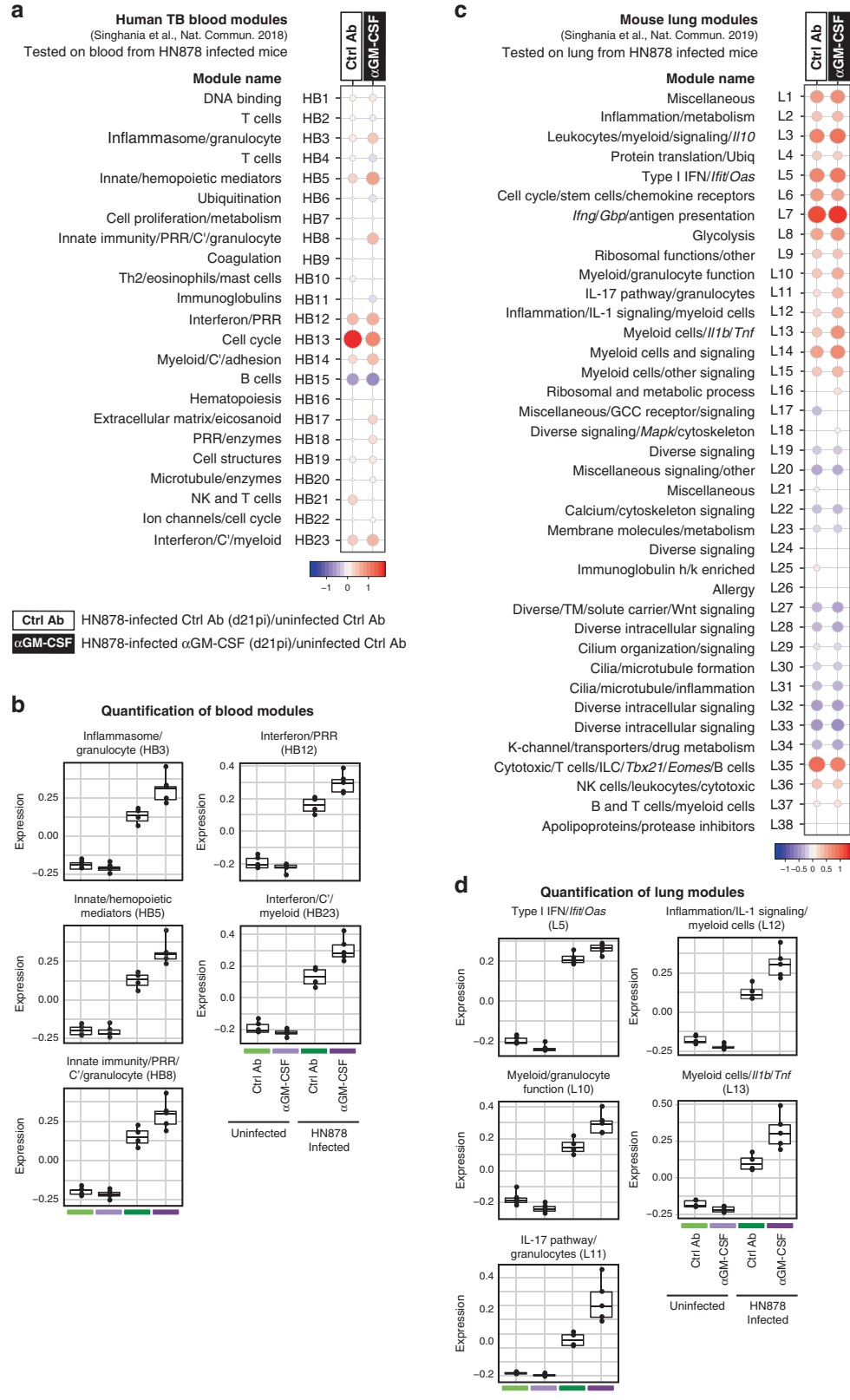

with TB resistance[16], were slightly less over-abundant in infected lungs from infected mice treated with αGM-CSF versus Ctrl Ab (Supplementary Data 5 and Fig. 2c). In all, the transcriptional profile of lungs from HN878-infected αGM-CSF treated mice resembled that of infected TB-susceptible mice[16].

**GM-CSF prevents NETosis during *M. tuberculosis* infection.** To further investigate the immune cell subsets associated with the exacerbated inflammation observed at the site of infection following GM-CSF blockade (Fig. 1d–f, Supplementary Data 1 and Fig. 2c), we tested our RNA-Seq lung data for enrichment of

**Fig. 2 GM-CSF controls transcriptional signatures associated with TB pathogenesis.** WT mice were infected and/or treated as in Fig. 1a. Blood and lungs were collected from *M. tuberculosis* HN878-infected and uninfected mice for RNA-Seq analysis (*n* = 4–5 mice/group). **a** Blood modules of co-expressed genes derived using WGCNA from human TB datasets[35] are shown for blood RNA-Seq datasets obtained from infected mice treated with αGM-CSF or Ctrl mAbs, compared to uninfected Ctrl Ab treated mice (Supplementary Data 3). **c** Lung modules of co-expressed genes derived using WGCNA from mouse lung disease modules[36] are shown for lung RNA-Seq datasets obtained from infected mice treated with αGM-CSF or Ctrl mAbs, compared to uninfected Ctrl Ab treated mice (Supplementary Data 5). **a**, **c** Fold enrichment scores derived using QuSAGE are depicted, with red and blue indicating modules over- or under-abundant, compared to the controls. Colour intensity and size of the dots represents the degree of perturbation, indicated by the colour scale, with the largest dot representing the highest degree of perturbation within the plot. Only modules with fold enrichment scores with FDR *P*-value < 0.05 were considered significant and depicted here. Module name indicates biological processes associated with the genes within the module. **b**, **d** Boxplots depicting the module eigengene expression, i.e., the first principal component for all genes within the module, are shown for uninfected Ctrl Ab (light green) or αGM-CSF (light purple) and infected Ctrl Ab (dark green) or αGM-CSF (dark purple) treated groups (*n* = 4–5 mice/group), for (**b**) blood modules: Inflammasome/Granulocyte (HB3), Innate/hemopoietic mediators (HB5), Innate Immunity/PRR/C'/Granulocyte (HB8), Interferon/PRR (HB12) and Interferon/C'/Myeloid (HB23); and (**d**) lung modules: Type I IFN/*Ifit/Oas* (L5), Myeloid/granulocyte function (L10), IL-17 pathway/granulocytes (L11), Inflammation/IL-1 signalling/myeloid cells (L12), and Myeloid cells/*Il1b/Tnf* (L13). Gene expression profiles are shown using boxplots, where the upper and lower box limits show the interquartile range (limits of second and third quartiles), a thick horizontal bar within the box shows the median, whiskers show the minimum and maximum values, and each dot represents an individual mouse (*n* = 4–5 mice/group). C', complement.; PRR, pathogen recognition receptor. GCC, glucocorticoid; K-channel, potassium channel; TM, transmembrane; Ubiq, ubiquitination.

previously curated immune cell-type associated gene sets[40]. Gene set enrichment analysis (GSEA) revealed that the neutrophil-associated gene set was the highest positively enriched set in infected lungs from αGM-CSF versus Ctrl Ab treated mice (Fig. 3a; normalized enrichment score (NES) = 6.8, FDR < 0.0001), in keeping with the increased over-abundance of neutrophil-associated modules in the blood (HB3, HB5 and HB8; Fig. 2b) and lung transcriptional signatures (L10 and L11; Fig. 2d). The expression of genes associated with neutrophil recruitment and activation, such as *S100a6*, *S100a8*, *S100a9*, *Mmp8*, *Mmp9*, *Cxcr2*, *Cd177* and *Lcn2*, was upregulated in infected lung samples but further increased in lungs from infected mice treated with αGM-CSF mAbs compared to infected Ctrl Ab treated mice (Fig. 3b and Supplementary Data 5). Increased expression of these neutrophil-associated genes was also observed in the blood from infected mice treated with αGM-CSF compared to infected Ctrl Ab mice (Supplementary Fig. 3b and Supplementary Data 3). Despite the increased expression of the majority of genes in the neutrophil-associated gene set driven by GM-CSF blockade (Fig. 3b), we observed a modest and variable increase in the number of neutrophils and inflammatory Ly6C$^+$ monocytes in the lungs from infected mice treated with αGM-CSF versus Ctrl Ab (Fig. 3c, Supplementary Fig. 5 and Supplementary Data 1), indicating that the neutrophil-associated transcriptional signature was resulting from activation and not only increased numbers of neutrophils. In contrast, alveolar macrophages were consistently decreased in the lungs from infected mice treated with αGM-CSF mAbs compared to infected Ctrl Ab treated mice (Fig. 3c, right panel). This is in line with the downregulation of genes associated with alveolar macrophages induced by GM-CSF blockade (Supplementary Data 4 and Supplementary Fig. 4c, right panel). To investigate the role of neutrophils in mycobacterial control we treated mice with mAbs against Ly6G from one-week post-infection. Notably, neutrophil depletion led to reduced mycobacterial growth in the lungs of infected mice treated with αGM-CSF mAbs (Fig. 3d), confirming a role for neutrophils in the disease exacerbation observed upon GM-CSF blockade during infection.

As neutrophils were required for disease exacerbation (Fig. 3d) and their activation profile by gene expression appeared altered following GM-CSF blockade during infection (Fig. 3b), we examined neutrophil responses in vivo by immunofluorescence microscopy. We found an increased accumulation of neutrophils, detected by myeloperoxidase staining (MPO$^+$), in the lesions of infected αGM-CSF treated mice compared to the lung lesions from infected Ctrl Ab treated mice (Fig. 3e). These findings prompted us to determine whether NETs were also present by examining the

colocalization of DAPI (DNA staining) and citrullinated H3 (CitH3), a specific marker for NET formation[41,42]. NETs were detected in surprisingly high amounts in the lung lesions from αGM-CSF treated mice but were scarcely found in infected Ctrl Ab treated mice, as shown quantitatively by measuring the percentage of CitH3 staining areas within the MPO positive areas in the lung lesions (Fig. 3f). These data indicate that neutrophil-driven disease exacerbation induced by GM-CSF blockade during *M. tuberculosis* infection correlates with excessive NETosis in the infected lungs.

**NETs are detected in TB-susceptible mice and human TB.** We have recently reported that over-abundance of neutrophil-associated genes in the blood and lungs also correlated with lung disease severity in other mouse models of increased susceptibility to TB, such as the highly susceptible C3HeB/FeJ mice infected with *M. tuberculosis* HN878[16]. However, the mechanisms of neutrophil-mediated pathology in these TB-susceptible mice remain unknown. Neutrophil depletion by the administration of mAbs against Ly6G after one-week post-infection, and then every other day, resulted in reduced mycobacterial loads in the lungs of *M. tuberculosis*-infected C3HeB/FeJ mice (Fig. 4a), confirming the deleterious role for neutrophils in host control of mycobacterial replication in C3HeB/FeJ mice as previously reported[19]. Given our findings of a high abundance of NETs at the site of infection upon blockade of GM-CSF, which correlated with increased blood and lung neutrophil-associated transcriptional signatures and disease severity, we hypothesised that NETosis also occurs in the lungs of TB-susceptible C3HeB/FeJ mice. We found that NETs (DAPI$^+$CitH3$^+$) were abundant in the neutrophilic (MPO$^+$) lung lesions from infected C3HeB/FeJ mice (Fig. 4b), showing that NET formation correlates with disease severity in other models of increased susceptibility to TB other than in *M. tuberculosis*-infected mice during GM-CSF blockade.

Importantly, we have also reported that over-abundance and increased expression of genes associated with activation and recruitment of neutrophils in the blood of active TB patients correlated with the extent of lung radiographic disease[16]. We therefore investigated the presence of NETs in lung lesions from a cohort of patients undergoing resection for pulmonary TB, as after completion of chemotherapy these patients had radiographic evidence of residual disease[43]. Notably, in 12 out of 13 lung sections examined (one section per TB patient, Supplementary Table 1), we detected NETs immediately bordering and within areas of caseous necrosis (Fig. 5 and Supplementary Fig. 6a), but

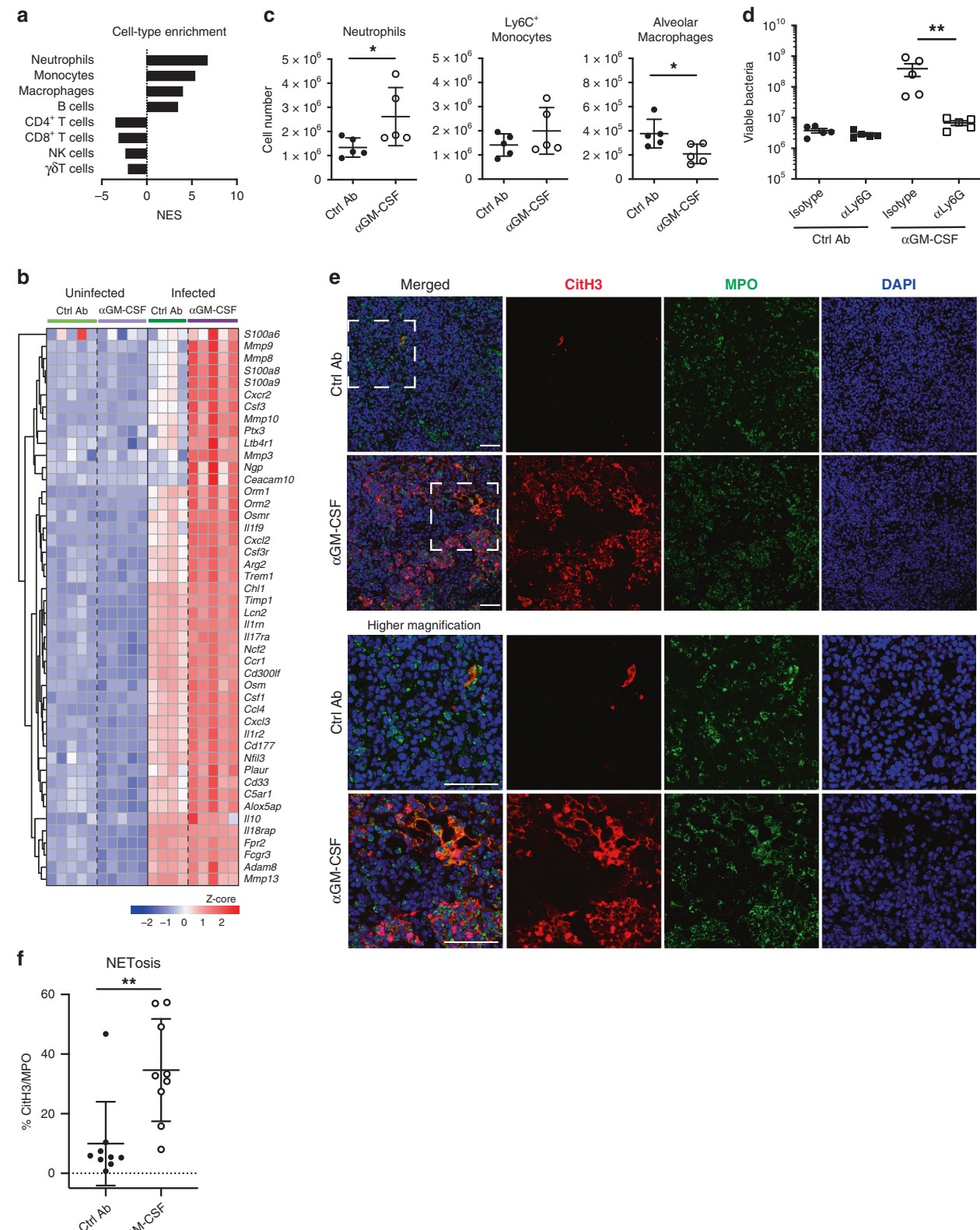

not outside the necrotic TB lesions. Consistent with the presence and potential contribution of neutrophils to the necrotizing lung lesions in human TB, analysis of protein content previously reported by Marakalala et al.[44] showed that neutrophil-associated proteins were enriched in the necrotic/caseous areas compared to the cellular periphery of human TB granulomas (Supplementary Fig. 6b). Our data thus demonstrate that NETs are formed in

human TB necrotic lesions showing a poor response to antibiotic therapy, in support of a role for NETs in human TB pathogenesis.

**Type I IFN signalling induces NETosis upon GM-CSF blockade.** To identify the factors leading to exacerbated NETosis during *M. tuberculosis* infection in the absence of GM-CSF signalling, we

**Fig. 3 Disease exacerbation upon GM-CSF blockade is neutrophil-dependent and correlates with NETosis. a** GSEA of immune cell-type associated gene sets[40] showing normalized enrichment scores (NES) for lung RNA-Seq data from infected αGM-CSF versus infected Ctrl Ab treated mice. **b** Heatmap showing relative expression of genes in the neutrophil-associated gene set for individual lung samples from uninfected Ctrl Ab (light green) or αGM-CSF (light purple) and infected Ctrl Ab (dark green) or αGM-CSF (dark purple) treated mice (n = 4–5 mice/group). Gene expression values were averaged and scaled across the row to indicate the number of standard deviations above (red) or below (blue) the mean, denoted as row Z-score. The dendrogram shows unsupervised hierarchical clustering of genes. **c** WT mice were infected and treated with Ctrl Ab (closed circles) or αGM-CSF (open circles) as in Fig. 1a. Lung cell suspensions were prepared and stained for the detection of myeloid cells (Supplementary Fig. 5). Cell numbers for neutrophils (CD11b+ Ly6G+), Ly6C+ monocytes (CD11b+Ly6G−Ly6C+) and alveolar macrophages (CD11b^lowCD11c+) are shown. *P = 0.0317. **d** WT mice were infected and treated with Ctrl Ab (closed symbols) or αGM-CSF (open symbols) as in Fig. 1a with the exception that from day 7 post-infection, mice also received 0.2 mg of anti-Ly6G (αLy6G; squares) or isotype control (labelled as Isotype; circles) mAbs every other day by i.p. injection. Lung viable bacterial loads were determined. **P = 0.0079. **e** Representative images of lung sections stained with citrullinated histone H3 (citH3; red), MPO (green), and DAPI (blue). NETs are visualized by colocalization of citH3 and DAPI staining (merged images; n = 3 mice/group). Bottom rows show detail from the top rows for each group. Scale bars, 50 μm. **f** Percentage of NET area in the lung normalized to MPO positive signal. **P = 0.0019. Data representative of one (**a**, **b**), two (**d**) or five (**c**, **e**) biological experiments; or data pooled of three independent experiments (**f**). Represented is the mean ± SD (**c**, **f**) or mean ± SEM (**d**); each dot represents an individual mouse: n = 5 mice/group (**c**, **d**) or n = 9 mice/group (**f**). Source data are provided as a Source Data file. Statistical analysis was performed using two-tailed Mann–Whitney test.

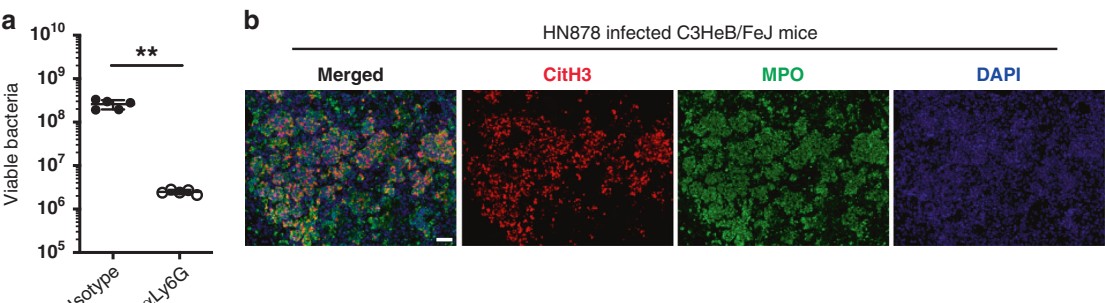

**Fig. 4 NETs are detected in lung lesions of TB-susceptible C3HeB/FeJ mice. a** C3HeB/FeJ mice were aerosol infected with *M. tuberculosis* HN878 treated by i.p. injection with 0.2 mg of anti-Ly6G (αLy6G; open circles) or isotype control (labelled as Isotype; closed circles) mAbs every other day, from day 7 post-infection until harvest on day 26 post-infection. Lung viable bacterial loads were determined. **P = 0.0079. Represented is the mean ± SD; each dot represents an individual mouse: n = 5 mice/group. Source data are provided as a Source Data file. Statistical analysis was performed using two-tailed Mann–Whitney test. **b** C3HeB/FeJ mice were aerosol infected with *M. tuberculosis* HN878 and their lungs harvested at day 35 post-infection. Formalin-fixed paraffin-embedded lung sections were stained with citrullinated histone H3 (citH3; red), MPO (green), and DAPI (blue). NETs are visualized by colocalization of citH3 and DAPI staining (merged image; n = 3 mice/group). Scale bars, 50 μm. Data from one experiment.

analysed the global transcriptional changes at the site of infection caused by αGM-CSF treatment. GSEA revealed significant enrichment of several hallmark gene sets in the lungs from αGM-CSF compared to infected Ctrl Ab treated mice (Supplementary Fig. 7a). Top positively enriched gene sets included biological processes associated with host resistance to *M. tuberculosis* infection[4,21,22,24], such as TNF, IFN-γ and IL-6 signalling (Supplementary Fig. 7a), which in excess can themselves be associated with pathology[45–48]. Interestingly, we also found genes associated with the IFN-α response among the top 5 positively enriched gene sets (Supplementary Fig. 7a–c), consistent with the increased over-abundance of the Type I IFN/*Ifit*/*Oas* module (L5) observed in the lungs from αGM-CSF treated compared to infected Ctrl Ab treated mice (Fig. 2c, d). Since increased and sustained levels of type I IFN signalling have been shown to exacerbate TB in several settings[10–15], we then tested whether the increased type I IFN-response, induced by GM-CSF blockade during *M. tuberculosis* infection, promoted disease exacerbation. Confirming this hypothesis, type I IFN receptor-deficient (*Ifnar*−/−) mice infected with HN878 and treated with αGM-CSF mAbs exhibited reduced lung mycobacterial loads compared to infected αGM-CSF treated WT mice (Fig. 6a). This reduced mycobacterial load was brought to the level detected in infected control WT and *Ifnar*−/− mice in the absence of αGM-CSF mAbs, indicating that type I IFN signalling is responsible for disease exacerbation upon GM-CSF blockade. Consistently, the exacerbated lung pathology prompted by αGM-CSF blockade during infection was also dependent on

type I IFN signalling (Fig. 6b–d and Supplementary Data 1). We observed less severe lesions, with reduced lung lesion burden (Fig. 6b, c) and reduced necrosis and intra-alveolar necrotic debris (Fig. 6d), which also correlated with decreased neutrophil infiltration (Supplementary Data 1), in the lungs from infected *Ifnar*−/− mice compared to WT mice treated with αGM-CSF mAbs. Of note, we also observed reduced lung necrosis (Fig. 6d) and neutrophil infiltration (Supplementary Data 1) in the absence of IFNAR in the infected Ctrl Ab groups, indicating that type I IFN signalling promotes neutrophil recruitment and necrosis during *M. tuberculosis* infection even in the setting of intact GM-CSF signalling. In contrast to WT mice, lesions containing multi acid-fast bacilli were not detected in the infected lungs from *Ifnar*−/− mice treated with αGM-CSF mAbs (Fig. 6e), in keeping with the reduced lung mycobacterial loads observed in the absence of IFNAR signalling during GM-CSF blockade in *M. tuberculosis*-infected mice (Fig. 6a). NETs were detected in abundance in the lung lesions from infected WT mice treated with αGM-CSF mAbs but practically non-existent in the lung lesions from *Ifnar*−/− mice, as shown quantitatively (Fig. 7a, b). These findings demonstrate that elevated levels of type I IFN signalling are required for exacerbated NETosis in the lung and disease severity triggered by GM-CSF blockade during *M. tuberculosis* infection.

**Neutrophil over-activation by type I IFN promotes TB.** We have previously shown that neutrophils isolated from patients

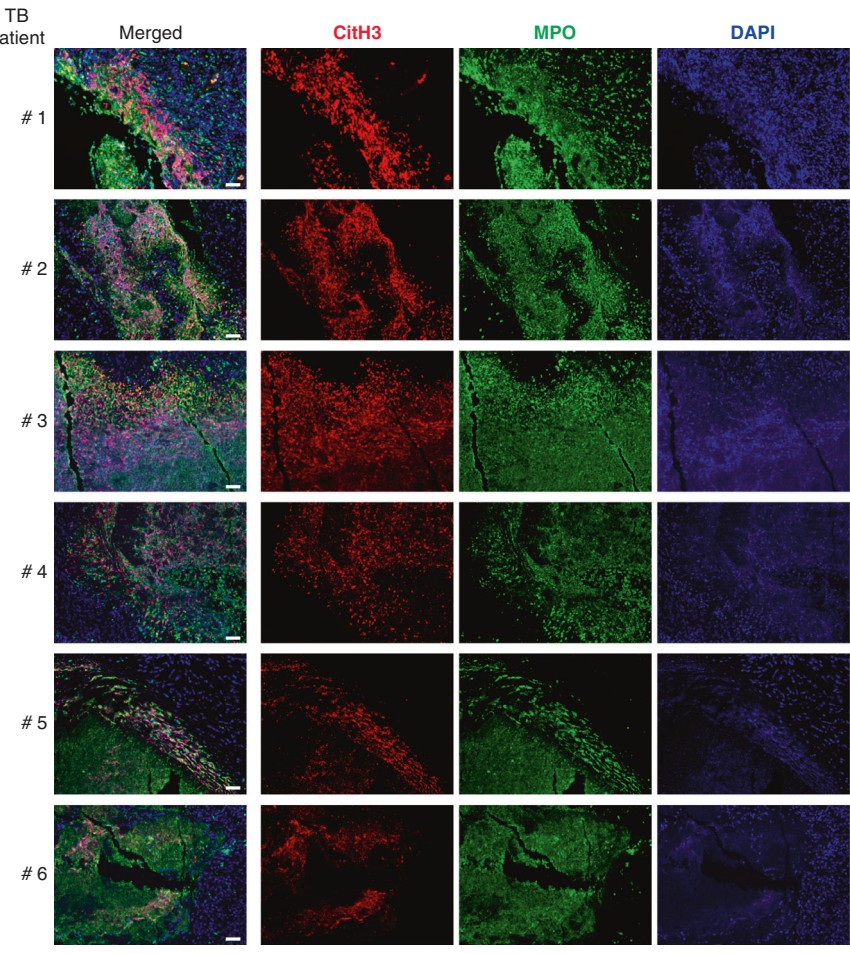

**Fig. 5 NETs are detected in necrotic lung lesions of human pulmonary TB.** Formalin-fixed paraffin-embedded lung sections from a total of thirteen patients with pulmonary TB (1 section per TB patient; Supplementary Table 1) were labelled with antibodies specific for citrullinated histone H3 (citH3; red) MPO (green) and DAPI (blue). NETs, visualized by colocalization of citH3 and DAPI staining (merged images, left), are shown for 6 TB patients. Scale bars, 50 μm. (All thirteen slides were stained and scanned together. See also Supplementary Fig. 6a).

with active TB over-express type I IFN-inducible genes[6,49], suggesting that over-activation of neutrophils by type I IFN during infection may contribute to disease progression. To determine whether type I IFN acts directly on neutrophils to promote disease severity during *M. tuberculosis* infection, we crossed *Ifnar*fl/fl mice with MRP8 (also known as S100a8) Cre mice to delete *Ifnar* in neutrophils[50,51]. Supporting our hypothesis, we observed reduced mycobacterial loads in the lungs from *M. tuberculosis*-infected *Ifnar*fl/fl-MRP8-Crepos mice, which lack *Ifnar* in neutrophils, compared to *Ifnar*fl/fl-MRP8-Creneg mice, both receiving treatment with αGM-CSF mAbs (Fig. 8a). Similar lung mycobacterial loads were detected between *Ifnar*fl/fl-MRP8-Crepos and *Ifnar*fl/fl-MRP8-Creneg infected mice treated with Ctrl Ab (Fig. 8a). NETs were detected in the lung lesions from infected *Ifnar*fl/fl-MRP8-Creneg and *Ifnar*fl/fl-MRP8-Crepos mice treated with αGM-CSF mAbs, but could barely be detected in infected mice treated with Ctrl Ab (Fig. 8b, c). A similar percentage of normalised NET area was quantified in the lung lesions from both groups of infected mice treated with αGM-CSF mAbs (Fig. 8c), suggesting that type I IFN is acting via other cells to promote NET-formation in neutrophils. However, the observation that lung neutrophil infiltration was found to be reduced in infected *Ifnar*fl/fl-MRP8-Crepos mice compared to *Ifnar*fl/fl-MRP8-Creneg mice treated with αGM-CSF mAbs (Fig. 8b), suggests that type I IFN signalling in neutrophils is promoting their recruitment to the site of infection. Collectively, these findings suggest that

over-activation of neutrophils by type I IFN impairs control of mycobacterial replication and promotes disease during *M. tuberculosis* infection, supporting a role for type I IFN signalling in neutrophil-driven TB pathogenesis.

**IFNAR signalling induces NETosis in TB-susceptible mice.** Next, we examined whether the dependence on type I IFN signalling for exacerbated NET formation and neutrophil-driven disease was more generally observed in models of increased susceptibility to TB that do not result from GM-CSF blockade. To address this, C3HeB/FeJ mice were treated with anti-IFNAR (αIFNAR) or isotype control (labelled as Ctrl Ab) mAbs the day prior to *M. tuberculosis* HN878 infection, and during the course of infection (Fig. 9a). Blockade of IFNAR signalling in susceptible C3HeB/FeJ mice infected with *M. tuberculosis* resulted in significantly reduced lung mycobacterial loads (Fig. 9b) and pathology (Fig. 9c), as compared to infected Ctrl Ab treated mice. Scoring of histological sections showed a significant reduction in lung lesion burden (Fig. 9d), necrosis and intra-alveolar necrotic debris (Fig. 9e), accompanied by fewer neutrophils (Supplementary Data 1), in infected C3HeB/FeJ mice treated with αIFNAR mAbs compared to Ctrl Ab treated mice. A reduced number of acid-fast bacilli were also detected in the lungs of infected C3HeB/FeJ mice upon IFNAR blockade (Fig. 9f). Notably, blockade of IFNAR signalling in susceptible C3HeB/FeJ mice abrogated NET formation at

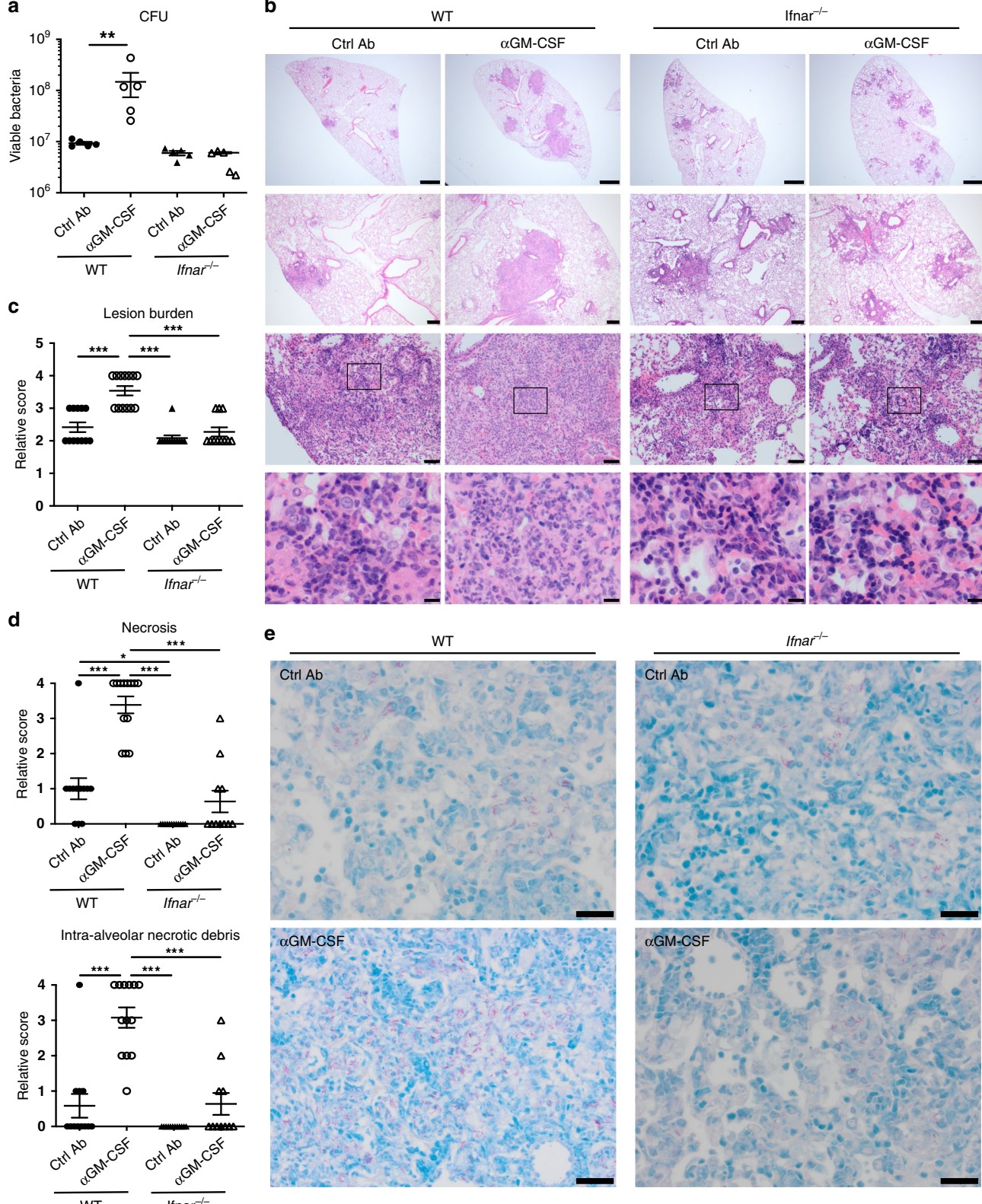

the site of infection, with significantly fewer NET-forming neutrophils detected in the lung lesions from αIFNAR as compared to Ctrl Ab treated mice, as shown quantitatively (Fig. 9g, h). These findings show that uncontrolled mycobacterial replication, exacerbated NETosis and pulmonary necrosis are also dependent on type I IFN signalling in TB-susceptible C3HeB/FeJ mice, showing that this is a generalised mechanism underlying TB pathogenesis.

## Discussion

GM-CSF has been increasingly appreciated as an essential mediator of protective immunity against TB[25–31]. However, the mechanisms underlying the pathology observed in the absence of GM-CSF signalling during *M. tuberculosis* infection remain poorly understood, particularly since past studies performed in GM-CSF-deficient mice had confounding lung pathologies[33,34],

**Fig. 6 IFNAR deletion abrogates disease exacerbation prompted by GM-CSF blockade during infection.** WT (circles) and *Ifnar*⁻ᐟ⁻ (triangles) mice were infected and treated with Ctrl Ab (closed symbols) or αGM-CSF (open symbols) as in Fig. 1a. **a** Viable bacterial loads determined in the lungs at day 21 post-infection. **\*\****P* < 0.0079. Represented is the mean ± SEM; each dot represents an individual mouse: *n* = 5 mice/group. Source data are provided as a Source Data file. Data representative of two biological experiments. Statistical analysis was performed using the two-tailed Mann–Whitney test. **b** Representative H&E staining of infected lung sections (*n* = 2–4 mice/group). Scale bars from top to bottom: 1 mm; 200 μm; 50 μm; 10 μm. **c** Relative lung lesion burden from H&E stained sections (Supplementary Data 1). 0 = no lesions, 1 = focal lesion, 2 = multiple focal lesions, 3 = one or more focal severe lesions, 4 = multiple focal lesions that are extensive and coalesce, 5 = extensive lesions that occupy the majority of the lung lobe. **\*\*\****P* < 0.0001. **d** Relative necrosis and intra-alveolar necrotic debris scores from H&E stained lung sections (Supplementary Data 1). 0 = not present, 1 = minimal, 2 = mild, 3 = moderate, 4 = marked. **\*\*\****P* < 0.0001; **\****P* = 0.0294. **c, d** Data pooled from four biological experiments. Represented is the mean ± SD; each dot represents an individual mouse: *n* = 11–13 mice/group. Source data are provided as a Source Data file. Statistical analysis was performed using one-way ANOVA Tukey's multiple comparisons test. **e** Representative Ziehl–Neelsen staining of infected lung sections (*n* = 2–4 mice/group). Scale bars, 20 μm. Data representative of four biological experiments (**b**, **e**).

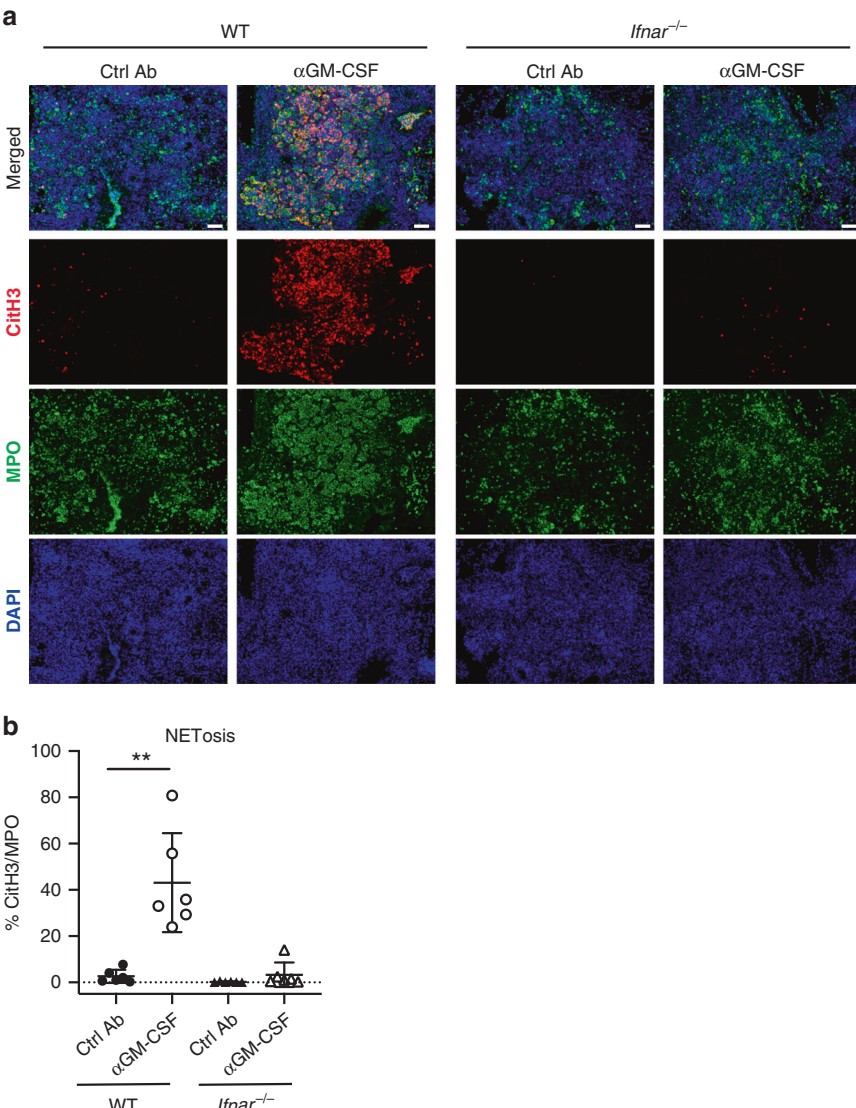

**Fig. 7 IFNAR deletion abrogates NET formation prompted by GM-CSF blockade during infection.** WT (circles) and *Ifnar*⁻ᐟ⁻ (triangles) mice were infected and treated with Ctrl Ab (closed symbols) or αGM-CSF (open symbols) as in Fig. 1a. **a** Representative images of lung sections stained with citrullinated histone H3 (citH3; red), MPO (green), and DAPI (blue). NETs are visualized by colocalization of citH3 and DAPI staining (merged images; *n* = 3 mice/group). Scale bars, 50 μm. Data representative of two biological experiments. **b** Percentage of NET area in the lung normalized to MPO positive signal. **\*\****P* = 0.0022. Data pooled from two biological experiments. Represented is the mean ± SD; each dot represents an individual mouse: *n* = 6 mice/group. Source data are provided as a Source Data file. Statistical analysis was performed using the two-tailed Mann–Whitney test.

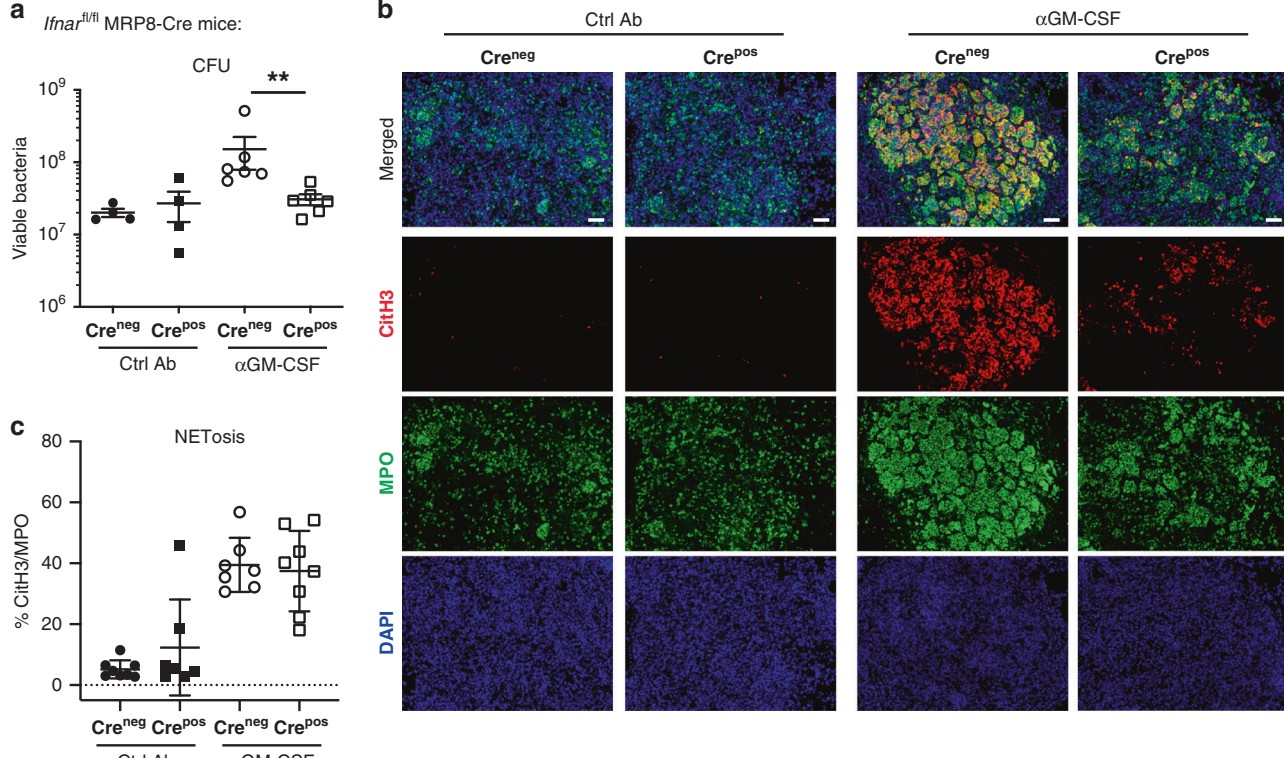

**Fig. 8 Type I IFN activates neutrophils to promote bacterial growth.** *Ifnar*^fl/fl MRP8-Cre negative (Cre^neg; circles) or positive (Cre^pos; squares) mice were infected and treated with Ctrl Ab (closed symbols) or αGM-CSF (open symbols) as in Fig. 1a. **a** Viable bacterial loads determined in the lungs at day 21 post-infection. **$P < 0.0022$. Data representative of two biological experiments. Represented is the mean ± SEM; each dot represents an individual mouse: $n = 4$–6 mice/group/experiment. **b** Representative images of lung sections stained with citrullinated histone H3 (citH3; red), MPO (green), and DAPI (blue). NETs are visualized by colocalization of citH3 and DAPI staining (merged images). Scale bars, 50 μm. Data representative of two biological experiments ($n = 3$–4 mice/group/experiment). **c** Percentage of NET area in the lung normalized to MPO positive signal. Data pooled from two biological experiments. Represented is the mean ± SD; each dot represents an individual mouse: $n = 7$–8 mice/group. Source data are provided as a Source Data file. Statistical analysis was performed using two-tailed Mann–Whitney test.

hampering interpretation of the findings. Here, using blocking mAbs, we describe an IFN-γ-independent mechanism of pathogenesis in the absence of GM-CSF signalling during *M. tuberculosis* infection. Transcriptomic analyses revealed that blockade of GM-CSF during *M. tuberculosis* infection results in increased expression of genes associated with neutrophil activation and a type I IFN response, which have been previously associated with disease severity in mouse models and human TB[16]. Confirming their role in TB pathogenesis, we show that disease exacerbation driven by either GM-CSF blockade during *M. tuberculosis* infection or during infection of susceptible C3HeB/FeJ mice was dependent on neutrophil over-activation by type I IFN leading to exacerbated NETosis at the site of infection. NETs were also found in necrotic lung lesions from patients with nonresolving pulmonary TB, further supporting a role for NETosis in TB pathogenesis.

Previous studies addressing the in vivo effects of GM-CSF on the immune response against *M. tuberculosis* infection have been performed using GM-CSF-deficient mice[27,31]. However, in the steady-state, these mice display dramatic defects on alveolar macrophages and lung function, including features of human PAP[33,34,38,39] and deficiencies in other immune cells[52,53]. This makes it difficult to distinguish the mechanisms of pathogenesis resulting from *M. tuberculosis* infection in the absence of GM-CSF from the developmental defects reported in the GM-CSF-deficient mice. Studies using GM-CSF-deficient mice[27,31] have suggested that GM-CSF is required for the recruitment of protective IFN-γ-producing T cells into the site of infection following

*M. tuberculosis* infection. Conversely, our findings showed that GM-CSF blockade during infection did not compromise IFN-γ-producing T cell numbers and instead resulted in increased transcriptional IFN-γ responses in infected lungs. Moreover, impaired control of mycobacterial growth driven by GM-CSF blockade during infection was also observed in the absence of IFN-γ signalling, revealing an IFN-γ-independent mechanism for the resulting disease exacerbation during *M. tuberculosis* infection in the absence of GM-CSF. Developmental defects in both alveolar macrophages and other immune cells such as dendritic cells[52,53], reported in GM-CSF-deficient mice, may account for the differences in T cell recruitment to the infected lungs observed in the GM-CSF-deficient mice[27,31], but not in the current scenario of lungs of adult mice where GM-CSF was only blocked during infection.

In contrast to the small impact of GM-CSF blockade on the numbers of T cells, neutrophils and monocytes, we found that GM-CSF blockade reduced the numbers of alveolar macrophages in the infected lungs, in keeping with the importance of GM-CSF in the maintenance and function of these cells[38,39]. It has been shown that the depletion of alveolar macrophages reduced *M. tuberculosis* burden in vivo[54], suggesting that alveolar macrophages are permissive to mycobacterial growth. Therefore, the reduced numbers of alveolar macrophages observed may not explain the increased *M. tuberculosis* burden observed upon GM-CSF blockade.

We found that GM-CSF blockade perturbed the expression of different networks of genes associated with protective versus

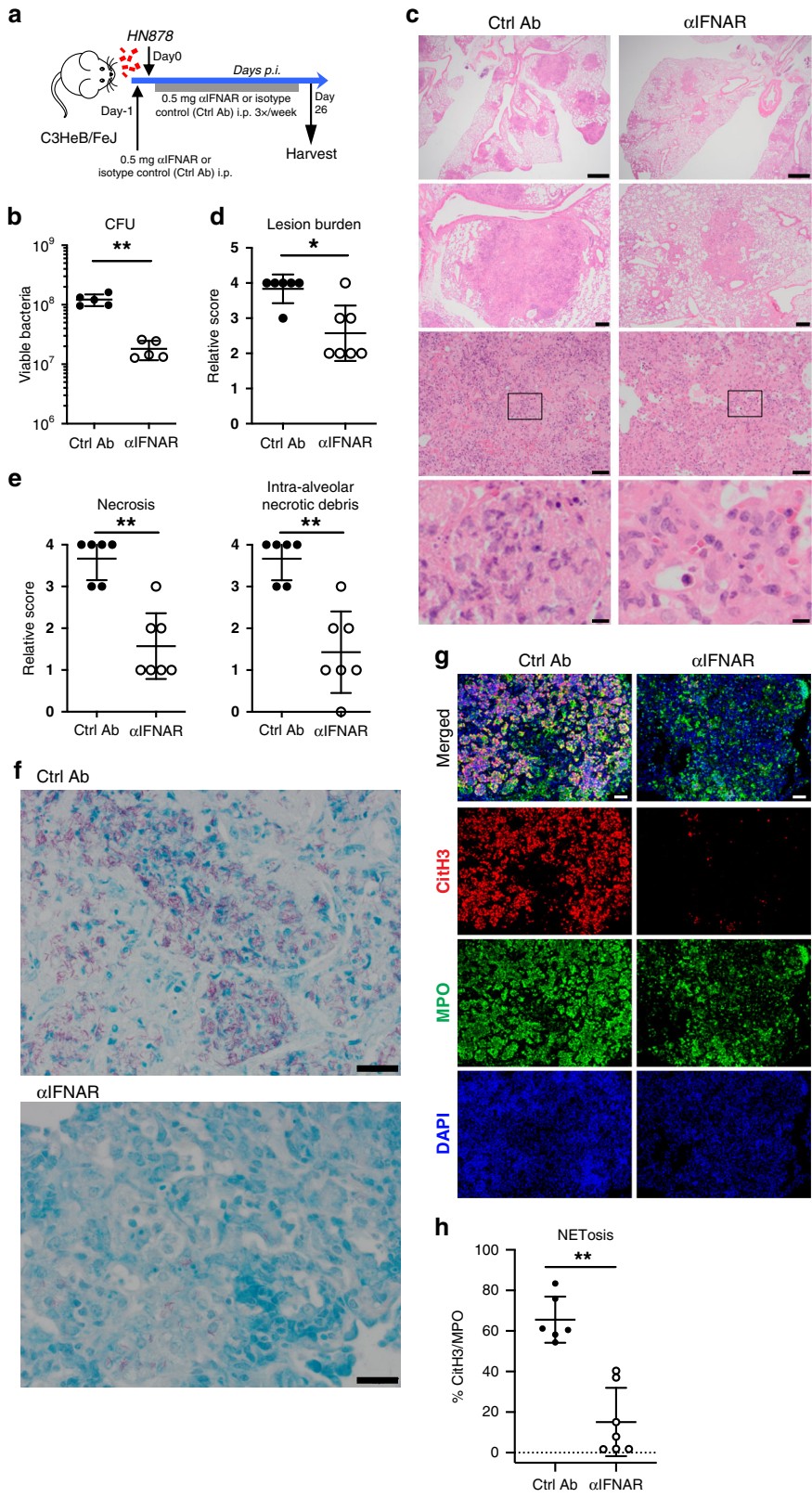

damaging immunity in the lungs and blood during *M. tuberculosis* infection. We have recently shown that transcriptional dominance of a type I IFN response and neutrophil activation, together with a loss of B cell, NK and T cell effector responses contribute to the pathogenesis of *M. tuberculosis* infection[16]. Using distinct but complementary transcriptomic analyses, here

we show that GM-CSF blockade boosted transcriptional responses reflective of type I IFN-inducible signalling and neutrophil activation during *M. tuberculosis* infection, promoting disease exacerbation. The effect of GM-CSF blockade to increase bacterial load was abrogated in the absence of IFNAR on neutrophils, providing an explanation to the link between type I IFN

**Fig. 9 IFNAR signalling is required for severity of lung NETosis in C3HeB/FeJ mice. a** Schematic of experimental design: C3HeB/FeJ mice were treated by intraperitoneal (i.p.) injection of 0.5 mg of anti-IFNAR (αIFNAR; open circles) or isotype control (labelled as Ctrl Ab; closed circles) mAbs the day prior low dose HN878 aerosol infection, followed by i.p. injections of 0.5 mg of αIFNAR or Ctrl mAbs, three times per weekly until harvest on day 26 post-infection. **b** Viable bacterial loads were determined. **P = 0.0079. **c** Representative hematoxylin and eosin (H&E) staining of infected lung sections (n = 3 mice/group). Scale bars from top to bottom: 1 mm; 200 μm; 50 μm, 10 μm. **d** Relative lung lesion burden from H&E stained sections (Supplementary Data 1). 0 = no lesions, 1 = focal lesion, 2 = multiple focal lesions, 3 = one or more focal severe lesions, 4 = multiple focal lesions that are extensive and coalesce, 5 = extensive lesions that occupy the majority of the lung lobe. *P = 0.0152. **e** Relative necrosis (left) and intra-alveolar necrotic debris (right) scores from H&E stained lung sections (Supplementary Data 1). 0 = not present, 1 = minimal, 2 = mild, 3 = moderate, 4 = marked. **P = 0.0023. **f** Representative Ziehl–Neelsen staining of infected lung sections. Scale bars, 20 μm. **g** Representative images of lung sections labelled with citrullinated histone H3 (citH3; red), MPO (green), and DAPI (blue). NETs are visualized by colocalization of citH3 and DAPI staining (merged images). Scale bars, 50 μm. **h** Percentage of NET area in the lung normalized to MPO positive signal. **P < 0.0012. **b**, **c**, **f**, **g** Data representative of two biological replicate experiments. **d**, **e**, **h** Data pooled from two biological experiments. Represented is the mean ± SD; each dot represents an individual mouse: n = 5 mice/group (**b**) or n = 6–7 mice/group (**d**, **e**, **h**). Source data are provided as a Source Data file. Statistical analysis was performed using the two-tailed Mann–Whitney test.

and neutrophil-driven transcriptional signatures. In contrast to the increased type I IFN and neutrophil-driven transcriptional signatures, GM-CSF blockade during infection, decreased the abundance of modules associated with NK, T and B cells, similarly to the decrease in these signatures seen in the blood of TB patients which correlated with advanced disease[16]. We also observed a decreased over-abundance in the blood Cell cycle module upon GM-CSF blockade during infection, which may be linked to the increased type I IFN signalling and the anti-proliferative effects of type I IFN[55].

Neutrophils have been emerging as key players of TB patho-genesis and their influx into the site of infection is associated with poor outcomes[16–20]. However, the mechanisms linking neu-trophils and increased TB susceptibilty are still unclear[56]. We observed enhanced neutrophil recruitment to the site of infection and NETosis following *M. tuberculosis* infection in the absence of GM-CSF signalling, in agreement with previous reports of increased neutrophilic infiltration into the lungs of GM-CSF-deficient mice following bacterial infection[57]. Transcriptional analysis revealed that the expression of genes associated with neutrophil recruitment but also activation was increased upon GM-CSF blockade in the blood and lungs from infected mice, suggesting that GM-CSF blockade during *M. tuberculosis* infec-tion affects not only the recruitment but also the activation status of neutrophils in circulation and at the site of infection, similarly to what was observed during infection of TB-susceptible C3HeB/FeJ mice[16]. We found that neutrophils were required for disease exacerbation induced in TB resistant C57Bl/6 mice by GM-CSF blockade during *M. tuberculosis* infection but additionally in TB-susceptible C3HeB/FeJ mice, and that neutrophil-driven disease exacerbation correlated with excessive NETosis at the site of infection in both models. NETs were originally described as a host antimicrobial defence mechanism against pathogens[41], but recent studies reported that NETs can modulate immune responses and may also induce damaging inflammation in var-ious infections and other inflammatory diseases[41,42,58–62]. NETs have been shown to be released in vitro by *M. tuberculosis*-infected mouse[63] and human[64,65] neutrophils, however, NETs did not prevent *M. tuberculosis* replication in vitro[63]. Consistently, the presence of the NET component citH3 in induced sputum[65] and neutrophil-related inflammatory proteins associated with NETs in sputum[65], serum[18,66,67] and lung biopsies[18,65] have been shown to reflect active TB and TB-related lung damage. We now show unequivocally that NETs are induced in human TB lung lesions showing a poor resolution after antibiotic therapy, further supporting a role for NETs in human TB pathogenesis and suggesting that although initially perceived as antimicrobials, NETs may promote TB infection and may limit the ability of

antibiotics to access the infected site. Although future mechanistic studies are required to fully decipher the contribution of NETosis to disease progression, these observations suggest that inhibiting NET release in conjunction with antibiotic therapy could be clinically beneficial and prevent long-term morbidity due to lung destruction in TB patients[68], particularly in cases of MDR or XDR TB where current anti-mycobacterial drugs are of limited use.

Our data are consistent with a critical role for type I IFN in the regulation of NETosis, and in turn exacerbated NETosis serving as a leading effector of type I IFN-induced pathology. In our infection models of increased susceptibility to TB, NETosis was completely abrogated in the absence of type I IFN signalling, which correlated with better control of infection and less tissue damage. Similarly, previous in vitro studies have reported that neutrophils produce NETs when exposed to plasma samples from patients with autoimmune disorders showing elevated levels of type I IFN or exogenous IFN-α[69,70]. These data suggested that type I IFN primes neutrophils for the formation of NETs. Several mechanisms have been proposed for the role of sustained and increased levels of type I IFN in exacerbating *M. tuberculosis* infection[7,10,11,13–15,55,71,72]. Our data indicate that this mechan-ism, that was originally identified in autoimmune and inflam-matory conditions, is also critically relevant in certain infection scenarios. Type I IFN appear to be key regulators of neutrophil antimicrobial strategy decisions in both sterile disease and infection, although the mechanisms may vary given that type I IFN signalling acts directly on neutrophils in autoimmune con-ditions. Although the contribution of type I IFN signalling in other cells to TB disease exacerbation has been previously reported[6,11,71,72], we now demonstrated that increased type I IFN signalling also promotes TB pathogenesis through direct stimu-lation of neutrophils in vivo during infection, leading to uncon-trolled mycobacterial growth at the site of infection. The selectivity of NET release in response to different microbes sug-gests that NETosis does not always result in an antimicrobial response, such as phagocytosis, to control certain pathogens[41], but that in other infections such as TB contributes to disease exacerbation.

Together, our data from TB patients and mouse models of increased susceptibility to TB by infection of susceptible C3HeB/FeJ mice, or resistant C57Bl/6 mice where GM-CSF signalling is abrogated during infection, indicate that excessive formation of NETs at the site of infection correlates with poor disease out-comes. Our findings provide an insight as to how uncontrolled neutrophil infiltration and activation may exert a detrimental effect during *M. tuberculosis* infection. Moreover, we show that this process is controlled by type I IFN signalling as demonstrated

by reduced NETosis and mycobacterial replication when IFNAR activity was abrogated by using IFNAR-deficient mice or anti-IFNAR mAbs during infection of TB-susceptible mice. These findings provide a mechanism to explain how the reported type I IFN-induced neutrophil signatures in human TB promote disease exacerbation.

## Methods

**Human samples**. The human samples used for this study came from the collection obtained within the SH-TBL project (Supplementary Table 1; https://doi.org/10.17632/knhvdbjv3r.1), led by the Experimental Tuberculosis Unit (UTE) and conducted in collaboration with the National Center for Tuberculosis and Lung Diseases of Georgia (NCTLD); and registered at the ClinicalTrials.gov database under code NCT02715271. The project was reviewed and approved by both the Ethics Committee of the NCTLD (IRB00007705 NCTLD Georgia #1, IORG0006411) and the Germans Trias I Pujol Hospital (IGTP) ethics committee (EC: PI-16-171). Written informed consent was obtained for the collection of biological material and data from all study participants before being enroled.

**Experimental animals and ethics**. C3HeB/FeJ, C57BL/6 wild-type (WT), IFN-γ receptor-deficient ($Ifngr1^{tm1Agt}$; IFNγR$^{-/-}$) and type I IFN receptor-deficient ($Ifnar1^{tm1Agt}$; $Ifnar^{-/-}$) mice were bred and housed in specific pathogen-free facilities at The Francis Crick Institute. $Ifnar^{flox/flox}$ ($Ifnar1^{tm1Uka}$; $Ifnar^{fl/fl}$) mice[73,74], which have loxP sites flanking exon 10 of the $Ifnar1$ gene, were crossed with MRP8-Cre (MRP8 also known as S100a8; Tg$^{(S100A8-cre,-EGFP)11Ilw}$) mice[50,51], to delete $Ifnar$ in neutrophils. $Ifnar^{fl/fl}$-MRP8-Cre$^{pos}$ mice and $Ifnar^{fl/fl}$-MRP8-Cre$^{neg}$ littermate control mice were used in experiments. All protocols for breeding were performed in accordance with Home Office (U.K.) requirements and the Animal Scientific Procedures Act, 1986. Age-matched females were used in experiments. Experiments were performed in accordance with Home Office (U.K.) requirements and the Animal Scientific Procedures Act, 1986, and with recommendations of the European Union Directive 2010/63/EU and approved by Portuguese National Authority for Animal Health – Direção Geral de Alimentação e Veterinária (DGAV-Ref.0421/000/000/2016). Mice were kept under specific-pathogen-free conditions, with controlled temperature (20–24 °C), humidity (45–65%) and light cycle (12-h light/dark). Mice were kept with food and water ad libitum and humanely euthanized by CO$_2$ asphyxiation.

**Experimental infection**. M. tuberculosis experiments were performed under ABSL-3 conditions. M. tuberculosis HN878 bacilli (clinical isolate) were grown to midlog phase in Middlebrook 7H9 broth supplemented with 10% oleic acid albumin dextrose complex (Difco), 0.05% Tween 80, and 0.5% glycerol before being quantified on 7H11 agar plates and stored in aliquots at −80 °C. Aliquots frozen at −80 °C were then thawed (6 aliquots) and quantified, to determine the concentration of the stored inocula. Mice were infected via the aerosol route using an inhalation exposure system, calibrated to deliver ~200 CFUs to the lung. The infection dose was confirmed by determining the number of viable bacteria in the lungs of five mice immediately after the aerosol infection. Infected mice were monitored regularly for signs of illness such as wasting, piloerection and hunching. Mice were euthanized by CO$_2$ inhalation and blood and/or lung samples from each group were collected from individual mice at day 21 post M. tuberculosis infection, unless otherwise indicated in the figure legend. Blood and/or lung samples from age-matched uninfected mice were collected at the same time and used as controls. Determination of lung bacterial load was performed by plating serial dilution of the organ homogenate on Middlebrook 7H11 agar supplemented with 10% oleic acid albumin dextrose complex plus PANTA to prevent contamination with other bacteria. CFUs were counted after 3 weeks of incubation at 37 °C, and the bacterial load per lung was calculated.

**Antibody-mediated GM-CSF blockade**. Anti-mouse-GM-CSF monoclonal antibody (αGM-CSF mAb: clone MP1-22E9, Lot A1102882) and isotype control mAb (rat IgG2a, clone GL117, Lot A1102882; labelled as Ctrl Ab throughout the manuscript and Figs. 1–3 and 6–8) were gifts from DNAX (now Merck, USA). Mice were injected intraperitoneally (i.p.) with 300 μl containing 1 mg of either αGM-CSF or isotype control mAbs diluted in PBS the day prior to M. tuberculosis infection. After infection, mice received twice weekly i.p. injections of 200 μL containing 0.3 mg αGM-CSF or isotype control mAbs for the course of the experiment. Similar results were obtained with mAbs purchased from BioXCell (αGM-CSF mAb: clone MP1-22E9, Lot 683718A1; isotype control mAb: rat IgG2a, clone 2A3, Lot 686318A2).

For IFN-γ blockade experiments, mice received 0.5 mg of either anti-mouse-IFN-γ (αIFN-γ) mAb (clone XMG1.2, Lot 5543-3/5543/0715; from BioXCell) or isotype control mAb (rat IgG1, clone GL113, Lot A2022916; gift from DNAX; labelled as Isotype in Supplementary Fig. 1d) together with αGM-CSF or isotype control mAbs for the course of the experiment.

**Antibody-mediated neutrophil depletion**. Anti-mouse-Ly6G (αLy6G) mAb (clone 1A8, Lot 695418J3) and isotype control (rat IgG2a, clone 2A3, Lot 71671801; labelled as Isotype in Figs. 3d and 4a) were purchased from BioXCell. M. tuberculosis-infected mice received a total of 7 i.p. injections containing 0.2 mg of αLy6G or isotype control mAbs every other day from day 7 post-infection.

**Antibody-mediated IFNAR blockade**. Anti-mouse-IFNAR1 (αIFNAR) mAb (clone MAR1-5A3, Lot 0419L555) was purchased from Leinco Technologies, Inc. Isotype control (rat IgG1, clone GL113, Lot A2022916; labelled as Ctrl Ab in Fig. 9) was a gift from DNAX (now Merck, USA). Mice were injected i.p. with 200 μL containing 0.5 mg of either αIFNAR or isotype control (Ctrl Ab) mAbs the day prior to M. tuberculosis infection and then three times per week after infection for the course of the experiment.

**Flow cytometry**. Single-cell homogenates prepared from harvested lungs from infected mice were stained according to manufacturer's instructions to exclude dead cells using a Live/Dead fixable red dead cell stain kit (Invitrogen). Cells were pre-treated for 10 min with anti-FcγRI/FcγRII (anti-CD16/CD32; clone 24G2, Harlan; 1:100) Ab and then stained with Abs against specific extracellular markers to identify myeloid cells or lymphocytes. Myeloid cell panel included Thy1.2 (clone 53-2.1, eBioscience; 1:400), Ly6G (clone 1A8, BD; 1:100), Ly6C (clone HK1.4, eBioscience; 1:200), CD11c (clone HL3, BD; 1:100), CD11b (clone M1/70, BD; 1:100). Lymphoid cell panel included Thy1.2 (clone 53-2.1, eBioscience; 1:400), CD3 (clone 145-2C11, eBioscience; 1:200), CD4 (clone RM4-5, eBioscience/BD; 1:200) and CD8 (clone 53-6.7, BD; clone Ly-2, eBioscience; 1:400). For detection of IFN-γ production, cells were restimulated ex vivo with M. tuberculosis tuberculin purified protein derivative (PPD; 20 μg/ml; Statens Serum Institute) and anti-CD28 (2 μg/ml; clone 37.51, Harlan) for 20 h. Brefeldin A (10 μg/ml; Sigma-Aldrich) was added during the last 4 h. After extracellular staining, cells were fixed and treated with permeabilization buffer (BD) according to manufacturer's instructions and stained with anti-IFN-γ (clone XMG1.2, eBioscience; 1:100) or isotype control (clone eBRG1, eBioscience; 1:100) Abs. All stained samples were fixed with stabilizing fixative (BD) and refrigerated in the dark overnight before being acquired on a CyAN ADP analyser (Dako, Ely, U.K.) using Summit software v4.4.0 (Cytomation). Data were analysed using FlowJo software v10.3 (Tree Star). Myeloid cell populations were analysed after exclusion of dead cells and Thy1.2$^+$ cells; neutrophils were gated as CD11b$^+$Ly6G$^+$, Ly6C$^+$ monocytes as CD11b$^+$Ly6G$^-$Ly6C$^+$ and alveolar macrophages as CD11b$^{low}$CD11c$^+$. T cell populations were analysed after the exclusion of dead cells and gating on Thy1.2$^+$CD3$^+$ cells.

**Histopathological analysis of lung samples**. Lung tissues from M. tuberculosis-infected mice were perfused and fixed in 10% neutral-buffered formalin followed by 70% ethanol, processed and embedded in paraffin, sectioned at 4 μm and stained with hematoxylin and eosin (H&E) or Ziehl-Neelsen (ZN). A single section from all lung lobes was viewed and scored as a consensus by three board-certified veterinary pathologists (S.L.P., E.H. and A.S-B.) blinded to the groups (Supplementary Data 1, which contains data from all original experiments contributing to the study). The relative lesion burden scoring (0–5 points) was determined using the following scale: 0 = no lesions, 1 = focal lesion, 2 = multiple focal lesions, 3 = one or more focal severe lesions, 4 = multiple focal lesions that are extensive and coalesce, and 5 = extensive lesions that occupy the majority of the lung lobe. A semi-quantitative scoring (0–4 points) method was devised to assess the following histological features: inflammatory cells (neutrophils, lymphocytes, macrophages), necrosis, and pleuritis; using the following scale: 0 = not present, 1 = minimal, 2 = mild, 3 = moderate, and 4 = marked changes. The presence of M. tuberculosis bacilli detected by ZN positive staining was scored as paucibacillary or multibacillary according to their abundance in the tissue. Representative images of each group were acquired on an Olympus BX43 microscope using an Olympus SC50 camera and cellSens Entry software (Ver. 1.18).

**Immunofluorescence detection of NETs**. To identify NETs from lung tissues, sections from M. tuberculosis-infected mice or from TB patients were de-waxed and re-hydrated before staining. Sections were treated with Dako Target Retrieval Solution, pH 9 Ready-to-Use (Agilent) at 97 °C for 45 min for antigen retrieval, and permeabilized in PBS 0.5% triton x-100. Samples were incubated with a blocking buffer (PBS with 2% BSA and 2% of donkey serum (Sigma-Aldrich)) for 1 h at room temperature (RT) and stained with rabbit anti-mouse antibodies directed against citrullinated histone H3 (ab5103 from Abcam; 1:500) and goat anti-mouse antibodies directed against MPO (AF3667, R&D systems; 1:40) overnight at 4 °C. After washing samples with PBS, sections were incubated with secondary donkey anti-rabbit IgG antibodies conjugated with Alexa Fluor 568 (A10042, Invitrogen; 1:200), donkey anti-goat IgG antibodies conjugated with Alexa Fluor 488 (A11045, Invitrogen; 1:200) and DAPI (Invitrogen; 1:1000) for 2 h in the dark at RT. Glass coverslips were mounted onto the slides using Prolong Gold (Invitrogen). Controls were stained with secondary antibody only, and nonspecific fluorescent staining was not detected when secondary antibodies were tested alone. Slides were analysed using Olympus VS120-L100 Slide Scanner and

the OLYMPUS OlyVIA software (v2.9). Representative images for each group were exported using QuPath software (v0.2.0-m5)[75].

The percentage of NET area normalized to MPO signal was calculated by exporting three regions of interest of 2000 × 2000 μm representative of each lung sections from Qupath (version 0.2.0) and analysed with FIJI (v 2.0.0). The images were split into the different channels; the DAPI channel was subjected to the threshold Triangle, a mask was generated to account for the total amount of tissue present in the region of interest. Default and Triangle thresholding was used with the MPO and the CitH3 channel respectively. The percentage of the area was calculated for the different channels based on the generated masks. The measured percentage of CitH3 signal was normalized to the measured percentage of MPO signal and the % CitH3/MPO values obtained from the three different regions were averaged and plotted. Statistical analysis of the plots was obtained by performing a nonparametric Mann–Whitney test.

**RNA extraction and pre-processing for RNA-Seq**. Blood was collected in Tempus reagent (Life Technologies) at 1:2 ratio. Total RNA was extracted using the Tempus™ Spin RNA isolation kit (Thermo Fisher Scientific). Globin RNA was depleted from total RNA (1.5–2 μg) using the GLOBINclear™-Mouse/Rat Globin mRNA removal kit (Thermo Fisher Scientific). Lungs were collected in TRI-Reagent (Sigma-Aldrich). Total RNA was extracted using the RiboPure™ Kit (Ambion). RNA quantity was verified using NanoDrop™ 1000/8000 spectrophotometers (Thermo Fisher Scientific). Quality and integrity of the total and the globin-reduced RNA were assessed using a BioAnalyzer 2100 (Agilent).

**RNA-Seq data analyses**. Biological replicate libraries were prepared from total/globin-reduced RNA (200 ng) using the KAPA mRNA polyA Hyper Prep kit (Illumina) and sequenced on an Illumina HiSeq 4000 sequencer platform generating ~30 million 101 bp paired-end reads per sample. RNA-seq data were subjected to quality control using FastQC and MultiQC. Fastq files were trimmed for Illumina adapter sequences using Trimmomatic (version 0.36) prior to alignment. The RSEM package (version 1.3.30)[76] in conjunction with the STAR alignment algorithm (version 2.5.2a)[77] was used for the mapping and subsequent gene-level counting of the sequenced reads with respect to Ensembl mouse GRCm.38.89 version genes. Normalisation of raw count data and differential expression analysis was performed with the DESeq2 package (version 1.18.1)[78] within the R programming environment (version 3.4.3)[79]. The top 500 genes with the highest variance across each normalised dataset were used to generate the PC analysis plot. The first 10 principal components association was tested against Infection (samples from HN878-infected versus uninfected mice) and αGM-CSF (samples from αGM-CSF versus isotype Ctrl mAbs treated mice) status using linear modelling. Barchart shows the % variance described by the first 10 PCs. The tiled plot shows the significance of each PC with αGM-CSF and Infection status as assessed by an F-test of the associated regression model. Differentially expressed genes were defined as those showing statistically significant differences, P values were adjusted for multiple testing using the Benjamini-Hochberg method (Supplementary Data 2 and 4). Volcano plots showing the logFC and adjusted P value (FDR) were plotted using ggplot2 (version 3.3.2) for differentially regulated genes (FDR < 0.05). Genes were colour coded based on up or downregulated relative to either uninfected Ctrl Ab or infected Ctrl Ab. Differentially expressed genes were plotted using pheatmap (version 1.0.12) using normalised log2 expression values, scaled per row indicating the standard deviation from the mean (Z score). Genes were clustered using a Euclidean distance matrix and clustered using Ward's hierarchical clustering algorithm[80]. Gene lists ranked by the Wald statistic were used to look for pathway and biological process enrichment using the Broad's GSEA software (version 2.1.0) with genesets from MSigDB (version 6)[81]. Gene lists ranked by the Wald Statistic comparing lung RNA-Seq data from infected αGM-CSF versus infected Ctrl Ab treated mice were also used to determine cell-type enrichment using cell-specific signatures previously curated[40] and GSEA. Bar plot shows the normalised enrichment score (NES) for specific cell signatures in the lungs from infected αGM-CSF versus infected Ctrl Ab treated mice, where the FDR < 0.05.

Cellular deconvolution analysis for quantification of relative levels of distinct cell types on a per sample basis was carried out on normalized counts using CIBERSORT[82]. CIBERSORT estimates the relative subsets of RNA transcripts using linear support vector regression. Mouse cell signatures for 25 cell types were obtained using ImmuCC and grouped into 9 representative cell types based on the application of ImmuCC cellular deconvolution analysis to the sorted cell RNA-seq samples from the ImmGen ULI RNA-seq dataset (ImmGen Consortium: GSE109125; http://www.immgen.org) as previously described[36].

For module enrichment analysis, the Bioconductor package qusage (version 2.18.0)[83] was used to determine the enrichment of gene sets identified from human TB blood datasets[35] and mouse lung disease datasets[36]. Fold enrichment results were plotted as a dotplot where red and blue colouring represents up and downregulation of cumulative abundance of module genes relative to controls. The size of the dots represents the relative change in cumulative abundance. Only modules with an adjusted P value < 0.05 are shown. Eigengene values per module were calculated on the VST normalised expression using the module Eigengenes function from WGCNA package.

**Reporting summary**. Further information on research design is available in the Nature Research Reporting Summary linked to this article.

## Data availability

RNA sequence data have been deposited in the NCBI Gene Expression Omnibus (GEO) database under the primary accession code GSE141207. Data regarding the sorted cells from Immunological Genome Project were accessed from GEO under the accession code GSE109125. All other data that support the findings of this study are available in the article and Supplementary Information files or from the corresponding author upon request. Source data are provided with this paper.

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

## Acknowledgements

We thank Xuemei Wu (Laboratory of Immunoregulation and Infection, The Francis Crick Institute) for her help in coordinating all the breeding and maintenance of the mice used in this study. We thank Jean Langhorne (Malaria Immunology Laboratory, The Francis Crick Institute) and Ilaria Malanchi (Tumour-Host Interaction Laboratory, The Francis Crick Institute) for the *Ifnar*fl/fl and the MRP8-Cre mice, respectively. We thank Akul Singhania and Olivier Tabone (Laboratory of Immunoregulation and Infection, The Francis Crick Institute) for critical discussion and input regarding RNA-Seq analyses. We thank Alan Sher (National Institute of Allergy and Infectious Diseases, NIH) and Paulo Vieira (Institut Pasteur) for discussion and critical reading of the manuscript. We thank The Francis Crick Institute Biological Services for animal husbandry and technical support; Advanced Sequencing Facility and Bioinformatics and Biostatistics Science Technology Platforms for helping with sequence sample processing and analyses; and Experimental Histopathology for their work in preparing lung sections for histological analyses. This study was funded by The Francis Crick Institute which receives its core funding from Cancer Research UK (FC001126, FC001999, FC001129), the UK Medical Research Council (FC001126, FC001999, FC001129), and the Wellcome Trust (FC001126, FC001999, FC001129); before that by the UK Medical Research Council (MRC U117565642); and by the European Research Council (294682-TB-PATH). The collection of human lung tissue samples for this study was funded by the Spanish Government-FEDER Funds through CP13/00174, CPII18/00031 and PI16/01511 grants, and the CIBER Enfermedades Respiratorias Network; and by the Spanish Society of Pneumology and Thoracic Surgery (SEPAR) through grant 16/023. A.O'G., L.M-T., P.J.S., E.S. and S.H. were supported by The Francis Crick Institute which receives its core funding from Cancer Research UK (FC001126), the UK Medical Research Council (FC001126), and the Wellcome Trust (FC001126); before that by the UK Medical Research Council (MRC U117565642). S.L.P., A.S-B., and E.H. were funded by the Royal Veterinary College and The Francis Crick Institute. M.S. was funded by grants POCI-01-0145-FEDER-028955, and by FCT through Estimulo Individual ao Emprego Científico. K.L.F. was funded by FCT PhD scholarship SFRH/BD/114405/2016.

## Author contributions

L.M-T. and A.O'G. conceived and designed the experiments. L.M-T and A.O'G. wrote the manuscript. M.S. managed some of the experiments in the study, and V.P. provided advice and technical help on neutrophil studies, and both provided feedback on data analysis and quantitation and helpful discussion for the project. L.M-T. performed most experiments, compiled the data and prepared the figures. E.S. assisted in all *M. tuberculosis* infection experiments. P.J.S. performed initial experiments for the project. S.H. contributed with experiments involving IFNγR$^{-/-}$ mice and anti-IFN-γ mAb treatment. J.S., K.L.F. and M.S. performed initial *M. tuberculosis* infection experiments with *Ifnar*$^{-/-}$ mice. P.C. analysed RNA-Seq data. E.H., S.L.P. and A.S-B. performed histopathological analysis and interpretation. M.I. and Q.W. carried out initial NETs staining and, together with V.P., contributed to the data interpretation. I.V.A performed quantification analysis and interpretation of NET-forming neutrophils in lung sections. S.V., P.R-M. and C.V. provided human lung tissue samples. All co-authors have read, reviewed, and approved the manuscript.

## Competing interests

The authors declare no competing interests.
