## [Peer Review File · Nature Communications]

REVIEWER COMMENTS

Reviewer #1 (Remarks to the Author):

GM-CSF has been implicated in enhancing host control of *M. tuberculosis* infection. The underlying mechanisms are poorly understood and functional studies in mouse models confounded by the physiological abnormalities of mice with genetic deletions in GM-CSF signalling. The study by Moreira-Teixeira et al. aims to overcome this shortcoming by using neutralising GM-CSF antibodies. The authors describe that anti-GM-CSF antibody administration exacerbates pathology and bacterial burden in lungs of C57Bl/6 mice infected with a highly virulent strain of *M. tuberculosis*, HN878. Administration of neutralising GM-CSF antibodies shifted blood and lung gene expression signatures towards profiles that have previously been associated with more severe TB disease in humans and in mouse models. GM-CSF neutralization was associated with enhanced neutrophil recruitment to infected lungs. This is intriguing and the opposite to what one might have expected based on previous studies where GM-CSF neutralization in various mouse models diminishes neutrophil infiltration into the lung, ameliorating inflammation. However, the authors might need to consider that prolonged neutralisation of GM-CSF during the acute phase of *M. tuberculosis* infection is likely to affect emergency myelopoiesis. The blood gene expression signatures in Suppl Fig 2c indeed indicate a shift towards neutrophil differentiation (possibly driven by G-CSF at the expense of monocyte differentiation) after 21 days of GM-CSF depletion. Neutrophilia induced by GM-CSF depletion might turn the relatively resistant C57BL/6 mouse into a more susceptible host more alike to mouse strains that show increased neutrophil infiltration and disease during *M. tuberculosis* infection (e.g. 129Sv). Analyses of blood neutrophil and monocyte counts in all experimental conditions should help assess to what extent the neutrophil/monocyte ratio is shifted and whether this correlates with neutrophil infiltration in the lung. Enhanced neutrophil infiltration in other infected organs (e.g. liver) might also be apparent.

In a series of *in vivo* experiments, the authors attribute the enhanced pathology and bacterial burden to neutrophil infiltration driven by type I IFN signalling specifically in neutrophils. Previous studies have established that neutrophil infiltration is associated with impaired *M. tuberculosis* control and linked type I IFN signalling to neutrophil-driven pathology. The finding that IFNAR signalling on neutrophils alone might promote recruitment to the lung is novel. In light of previously established links between IFN signalling and neutrophil recruitment (e.g. IL-1, 12/15-lipoxygenase, NO), it is worth expanding these studies to define which IFN-mediated mechanism drives the neutrophil recruitment into the lung in this model. Impact of abolished IFN signalling on circulating neutrophil numbers should be included to establish whether amelioration of neutrophil influx into the lung is associated with potential consequences for granulopoiesis.

In lung lesions of mice treated with anti-GM-CSF, the authors demonstrate dramatically enhanced staining for citrullinated histone 3, a marker used to identify NETs usually in combination with additional markers such as MPO or NE. Analyses in samples from patients with severe, unresolved TB show similar associations. The magnification of the immune fluorescence images is unfortunately too low to appreciate the full extent of overlapping Cit3H and MPO staining for the reliable identification of NETs in tissue sections. Quantification of the co-localization of molecular markers for NETs (e.g. as described by Brinkmann et al. 2016 *Front Immunol*) could be helpful. While there is an association between neutrophil infiltration, Cit3H staining and enhanced pathology, the data do not establish direct causality between type I IFN signalling in neutrophils and NET formation. Alternative scenarios of drivers of NETosis may include increased neutrophil density in lung lesions (facilitated by type I IFN signalling) leading to elevated local ROS production or local hypoxia enhancing neutrophil elastase activity.

The statistical analyses seem appropriate and the level of detail provided appears sufficient for others to independently reproduce these studies.

Other comments

1. Due to the small size of the figures, cells harbouring acid fast bacilli cannot be identified.
2. The MPO staining indicates a significantly different structure of the lung lesions in mice treated with anti-GM-CSF. Higher magnification analyses of the H&E stained sections should help identify whether this is representative of high density of intact neutrophils, neutrophils undergoing NETosis, or cellular debris in areas of necrosis.
3. Gene expression profiling analyses were done at one time point, 3 weeks post infection where clear differences in bacterial burden and pathology were observable and mice showed significant weight loss. Are there fundamental differences in blood and tissue gene expression, (neutrophil) infiltration, CFU, between control and anti-GM-CSF treatment earlier during infection? Do these correlate with the neutrophil/monocyte ratios in the blood?
4. Comparison with an *M. tuberculosis* isolate that induces lower type I IFN responses (e.g. CDC1551) might help further underpin the contributions of type I IFN to neutrophil infiltration in this model.

Reviewer #2 (Remarks to the Author):

The O'Garra laboratory were the first to highlight the importance of blood gene expression biosignatures in human with expression of Type 1 interferons and genes associated with neutrophils. Here they have investigated the role of GM-CSF in mouse models of tuberculosis and in resected lung tissue from TB patients. Giving a blocking antibody to GM-CSF prevented any developmental issues in GM-CSF knock-out mice. Interestingly the authors did not find that blocking GM-CSF significantly affected T cell numbers or expression of IFN γ , previously hypothesised as a mechanism through which GM-CSF might act. Blocking GM-CSF increased the TB related transcriptional signature previously identified in humans.

Having demonstrated the involvement of Type I interferons, and genes associated with neutrophils in mice, biopsies from a group of patients with TB were studied, showing that NETs were present in the lung tissues. The TB patients studied were patients having lung resections at the end of their chemotherapy as they had refractory disease. To be able to study any human lung tissue, rather than just blood, in patients with tuberculosis is really valuable. However these were patients who were having lung resections as they had refractory disease. Did they have drug-resistant TB? Although the Abstract states that "NETs are also detected in necrotic lung lesions of TB patients showing a poor response to antibiotic therapy" the interpretation that this suggests "a broad role in TB pathogenesis" is rather exaggerated. A role for PMN associated NETs in TB has been proposed by others (see refs 62-66) but this is to my knowledge the first demonstration of NETs in TB lung tissue.

It may also be worth referencing a study published in 2017 in *Front Immunol* by Guimaraes-Costa et al, that showed that NETs treatment of IL-4 and GM-CSF treated monocytes affected their ability to deal with *Leishmania* parasites.

Minor comments

Line 95. Delete extra space and full stop.

Line 189-. Please clarify here that this analysis was performed in infected mice.

Reviewer #3 (Remarks to the Author):

In this paper, Moreira-Teixeira et al seek to understand the contribution of GM-CSF signaling in the murine model of tuberculosis in a setting where alveolar macrophage development is not immediately disrupted. In this study, they utilized antibody-depletion to neutralize GM-CSF in vivo and compare this to animals treated with an isotype antibody. Nicely, the authors find similar results where loss of GM-CSF signaling enhances Mtb burden. The authors nicely demonstrate that this phenotype is independent of IFN γ R and IFN γ signaling. The authors next perform transcriptional analyses of the blood and the lung and compare these results to other publicly available datasets previously established in their groups. This drives the authors to hypothesize a role for neutrophils in the absence of GMCSF signaling. They then validate that treatment with a GMCSF antibody increases neutrophils as well as a decrease in alveolar macrophages. They nicely characterize neutrophil phenotype using the CitH3 antibody as a marker for netosis and see neutrophils have an altered phenotype during anti-GMCSF therapy. They see similar results in tissues obtained from humans with active TB undergoing surgery. They then further mine their data to implicate IFN signaling and show that loss of type I IFN receptor abrogates the altered bacterial burden observed in mice treated with the GMCSF antibody. They then show that deleting IFNAR in neutrophils results ameliorates the GMCSF phenotype.

In general, I highly enjoyed the paper and think the results are quite interesting. I think there are a few places where I think the authors could be a little more clear about their methodology.

Minor: In referencing their experiments involving an isotype control, I think it would be helpful to say label the Ctrl as Isotype Ab for clarity.

Minor: In Figure 1 e and 1f, is there an appropriate way to quantify the results statistically? Same for 6d.

Minor: In Figure 1g, the figure doesn't adequately convey the message I think the authors are trying to make.

Minor — Lines 189 through 194 are a bit difficult to study methodologically. Are these lung homogenates? Specific cells from the lung? I had to go to the methods to understand this experiment, and think the authors could be a little more clear in the main text.

Minor: Was there anything else different about the neutrophils that the authors would like to comment on from their transcriptional analysis?

Major: Is there any quantification that the authors can do to characterize their histopathology results? It's a little hard to think about changes in neutrophil abundance and state simultaneously.

Major: Are there any other tissues/samples that the authors have access to that show the robustness of their antibody?

Major: The authors make the jump from a transcriptional signature about IFN signaling in the lung without any protein validation for evidence of enhanced production of the cytokine during GMCSF antibody treatment. Did the authors see any evidence of this that they can comment on?

Major: While I recognize the power of their transcriptional signatures that they wanted to compare to, I think the authors missed an opportunity to speak more broadly about the transcriptomic data before transitioning to the analysis of the signatures. For example, I think they could have brought a little bit more of Supplemental Figure 2 into the main text to talk about the results more broadly.

REVIEWER COMMENTS

Response to Reviewer #1:

We thank the Reviewer for reading the manuscript and providing detailed feedback. We provide below answers to all comments raised by the Reviewer and have made changes to the manuscript with provision of additional data as detailed below. We believe that the revised version has resulted in a much improved manuscript.

Reviewer #1

“GM-CSF has been implicated in enhancing host control of *M. tuberculosis* infection. The underlying mechanisms are poorly understood and functional studies in mouse models confounded by the physiological abnormalities of mice with genetic deletions in GM-CSF signalling. The study by Moreira-Teixeira et al. aims to overcome this shortcoming by using neutralising GM-CSF antibodies. The authors describe that anti-GM-CSF antibody administration exacerbates pathology and bacterial burden in lungs of C57Bl/6 mice infected with a highly virulent strain of *M. tuberculosis*, HN878. Administration of neutralising GM-CSF antibodies shifted blood and lung gene expression signatures towards profiles that have previously been associated with more severe TB disease in humans and in mouse models. GM-CSF neutralization was associated with enhanced neutrophil recruitment to infected lungs. This is intriguing and the opposite to what one might have expected based on previous studies where GM-CSF neutralization in various mouse models diminishes neutrophil infiltration into the lung, ameliorating inflammation.”

Answer:

We would like to clarify that the enhanced neutrophil recruitment and exacerbated inflammation at the site of infection, observed in our model following GM-CSF neutralization during *M. tuberculosis* infection, has been reported before in other infection models. Our findings are consistent with previous reports in mice deficient in GM-CSF, showing exacerbated lung inflammation following *M. tuberculosis* infection: Gonzalez-Juarrero *et al.*, *J Leukoc Biol* 2005 [Ref. 25 in our revised manuscript]; Szeliga *et al.*, *Tuberculosis* 2008 [Ref. 29 in our revised manuscript]; and increased neutrophilic infiltration into the lungs following other bacterial infections (LeVine *et al.*, *J Clin Invest* 1999 [now included as Ref. 57 in our revised manuscript]). Moreover, it has been shown that local delivery of GM-CSF into the lungs reduced neutrophil infiltration by increasing GM-CSF availability into the lung during *Streptococcus pneumoniae* infection (Steinwede *et al.*, *Jl* 2011).

Previous studies that GM-CSF neutralization diminishes neutrophil infiltration into the lung, thus ameliorating inflammation, have indeed been reported, but these studies have used mouse models of sterile inflammation: e.g., by administration of LPS [Bozinovski *et al.*, *JBC* 2002] or inflammatory cytokines [Laan *et al.*, *Eur Respir J* 2003].

We have now clarified our referral to past reports, in our revised manuscript (page 5: lines 234-235; pages 20: lines 860-863 in the revised manuscript).

Reviewer #1

“However, the authors might need to consider that prolonged neutralisation of GM-CSF during the acute phase of *M. tuberculosis* infection is likely to affect emergency myelopoiesis. The blood gene expression signatures in Suppl Fig 2c indeed indicate a shift towards neutrophil differentiation (possibly driven by G-CSF at the expense of monocyte differentiation) after 21 days of GM-CSF depletion. Neutrophilia induced by GM-CSF depletion might turn the relatively resistant C57BL/6 mouse into a more susceptible host more alike to mouse strains that show increased neutrophil infiltration and disease during *M. tuberculosis* infection (e.g. 129Sv). Analyses of blood neutrophil and monocyte counts in all experimental conditions should help assess to what extent the neutrophil/monocyte ratio is shifted and whether this correlates with neutrophil infiltration in the lung. Enhanced neutrophil infiltration in other infected organs (e.g. liver) might also be apparent.”

Answer:

Here we address the Reviewer’s concern that prolonged neutralisation of GM-CSF during *M. tuberculosis* infection may “affect myelopoiesis by increasing the ratio of neutrophils to monocytes in the blood”.

It has been extensively reported that GM-CSF is totally dispensable for myelopoiesis at steady-state, and that abrogation of GM-CSF signalling has little to no effect on the neutrophil/monocyte ratio during “emergency myelopoiesis” induced by infection or other insults:

- Stanley *et al.*, PNAS 1994: Steady State showed no effect of GM-CSF on myelopoiesis [Ref. 31 in our revised manuscript];
- Nishinakamura *et al.*, Blood 1996: During steady state; or *Listeria monocytogenes* infection or administration of the cytotoxic drug, 5-fluorouracil, myelopoiesis was shown not to be affected;
- Griseri *et al.*, Immunity 2015: Colitis model, myelopoiesis was shown not to be affected.
- Zhan and Cheers, Blood 1998: Although they reported that in the absence of GM-CSF there was slightly lower emergency myelopoiesis during *Listeria* infection, these effects were variable; importantly in the context of our findings, the ratios of neutrophils to monocytes in their study were barely affected, and if anything favoured the monocyte lineage;
- Zhan and Cheers, Immunol Cell Biol 2000: In a mouse model of mycobacterial infection the same group, reported similarly that although the magnitude of the global myeloid response was slightly lower in the absence of GM-CSF, the neutrophil/monocyte ratio was not affected.

Thus, there is a body of literature conclusively showing that absence of GM-CSF does not lead to increased neutrophil production at the expense of monocytes, even during conditions of emergency myelopoiesis, secondary to infections. We would have liked to clarify our referral to these past reports, in our revised manuscript, however we are concerned that we are at the limit of the references allowed by *Nature Communications*, and we believe that this is out of the scope of our paper, particularly in view of the added data to the revised manuscript (detailed below).

Consistent with the above reports, in our model of *M. tuberculosis* HN878 infection of C57BL/6J mice, we see little preference for an increase in circulating neutrophils over other myeloid (monocytes/macrophages) cells in the presence of neutralizing anti-GM-CSF mAbs, as we show now using cellular deconvolution analysis of our blood RNA-Seq transcriptomic data (new data, new Supplementary Figure 3a in the revised manuscript), a technique which has previously been shown to mirror changes in cellular composition detected by flow cytometry (Singhania *et al.*, Nat Commun 2018). We show here that a similar immune cell composition was observed in uninfected mice treated with anti-GM-CSF (aGM-CSF) or isotype control (ctrl) mAbs for 3 weeks, in agreement with the dispensable role of GM-CSF for normal myelopoieses. During *M. tuberculosis* infection we observed an increase in the abundance of neutrophils in 3 out of 5 blood samples but this was accompanied by an increase in the overall abundance of all other myeloid (monocytes and macrophages) cells in all blood samples obtained from infected mice treated with aGM-CSF compared to ctrl mAbs (new Supplementary Figure 3a). Transcriptomic analysis of the blood samples, as the Reviewer referred to, showed that expression of genes not only associated with neutrophils (new Supplementary Figure 3b) but also monocytes (new Supplementary Figure 3c) and macrophages (new Supplementary Figure 3d) as we now point out more clearly, were increased upon GM-CSF blockade in *M. tuberculosis* infected mice, showing that all of these populations increased in the blood in the absence of GM-CSF. We apologise that this was not made clearer in the earlier version of our manuscript.

It is possible that by only highlighting the neutrophil-associated genes in the original Sup. Fig. 2c of our manuscript, we misled the Reviewer to think that GM-CSF blockade caused a shift towards neutrophil differentiation at the expense of monocytes and other myeloid cells during *M. tuberculosis* infection. We have now included the new cellular deconvolution data and heatmaps described above in our new Supplementary Figure 3 of our revised manuscript, to clarify this.

To support that our findings demonstrating that type I IFN-induced NETosis is a generalised mechanism of TB pathogenesis and not just observed upon neutralisation of GM-CSF in the context of *M. tuberculosis* infection, we now provide additional new data (new Figure 4 and new Figure 9) showing type I IFN-induced NETosis in TB-susceptible C3HeB/FeJ mice, which we have published recently show a blood signature resembling that of human TB (Moreira-Teixeira, O'Garra *et al.*, Nat Immunol 2020). In the revised manuscript, we now show that neutrophils are required for disease exacerbation in TB-susceptible C3HeB/FeJ mice during *M. tuberculosis* infection (new Figure 4a) and that neutrophil-driven disease exacerbation also correlated with excessive NETosis at the site of infection in this other model of increased susceptibility to TB (new Figure 4b). Moreover, we show that blockade of IFNAR signalling in TB-susceptible C3HeB/FeJ mice infected with *M. tuberculosis* HN878 (new Figure 9a) resulted in significantly reduced lung mycobacterial loads (new Figure 9b and f), reduced lung pathology (new Figure 9c-e), and abrogation of NET formation (new Figure 9g and h), as compared to infected control C3HeB/FeJ mice. This new Figure 9 shows that type I IFN-induced NETosis occurs in TB-susceptible C3HeB/FeJ mice infected with the clinical isolate of *M. tuberculosis* HN878, even in the absence of anti-GM-CSF neutralising antibodies, showing that this is a more generalised mechanism of TB pathogenesis over-and-above GM-CSF blockade. We have also revised the title, abstract and text throughout our manuscript to reflect these new

findings of a global role for type I IFN induced NET formation during *M. tuberculosis* infection of susceptible mice.

Reviewer #1

“In a series of in vivo experiments, the authors attribute the enhanced pathology and bacterial burden to neutrophil infiltration driven by type I IFN signalling specifically in neutrophils. Previous studies have established that neutrophil infiltration is associated with impaired *M. tuberculosis* control and linked type I IFN signalling to neutrophil-driven pathology. The finding that IFNAR signalling on neutrophils alone might promote recruitment to the lung is novel. In light of previously established links between IFN signalling and neutrophil recruitment (e.g. IL-1, 12/15-lipoxygenase, NO), it is worth expanding these studies to define which IFN-mediated mechanism drives the neutrophil recruitment into the lung in this model. Impact of abolished IFN signalling on circulating neutrophil numbers should be included to establish whether amelioration of neutrophil influx into the lung is associated with potential consequences for granulopoiesis.”

Answer:

We thank the Reviewer for recognising the novelty of our findings showing that type I IFN signalling on neutrophils promotes recruitment into the infected lungs and impairs control of mycobacteria replication. Our results also indicate that type I IFN signalling in other cells is also contributing to disease exacerbation by promoting NET formation during *M. tuberculosis* infection of resistant C57Bl/6 mice in the absence of GM-CSF signalling, and also in TB-susceptible C3HeB/FeJ mice. It is now well established in the literature that NETs are directly proinflammatory via their histones and DNA. For instance, it has been shown that IL-1b induction in a pulmonary fungal infection is decreased in transgenic mice expressing an EGFP-tagged H2B fusion protein, which block histone-mediated inflammation (Tsourouktsoglou *et al.*, Cell Reports 2020). In both of our models of type I IFN-dependent increased susceptibility to TB, we observed a direct correlation between the expression of *Il1b*, *Cxcl1* and *Cxcl2* and NET formation in the infected lungs (Figure 3b and Supplementary Data 4 for *M. tuberculosis* infection of resistant C57Bl/6 mice in absence of GM-CSF signalling; and Moreira-Teixeira *et al.*, Nat Immunol 2020 for *M. tuberculosis* infection of TB-susceptible C3HeB/FeJ mice). Therefore, it is likely that type I IFN-induced NETs might propagate inflammation and neutrophil recruitment in these models. Although of interest, the study of the mechanisms involved in neutrophil recruitment during *M. tuberculosis* infection is out of the scope of the current study.

Reviewer #1

“In lung lesions of mice treated with anti-GM-CSF, the authors demonstrate dramatically enhanced staining for citrullinated histone 3, a marker used to identify NETs usually in combination with additional markers such as MPO or NE. Analyses in samples from patients with severe, unresolved TB show similar associations. The magnification of the immune fluorescence images is unfortunately too low to appreciate the full extent of overlapping Cit3H and MPO staining for the reliable identification of NETs in tissue sections. Quantification of the co-localization of molecular markers for NETs (e.g. as described by Brinkmann *et al.* 2016 Front Immunol) could be helpful.”

Answer:

We thank the Reviewer for bringing this to our attention. We have now included a higher magnification in new Figure 3e of our revised manuscript to show the full extent of overlapping CitH3 and MPO staining for the identification of NETs in the lung sections. We have also quantified NETs as a percent of total neutrophils in the revised version of the manuscript by measuring the percentage of CitH3-positive stained area normalized to MPO-positive stained area for individual lung sections of each independent experiment (new Figures 3f, new Figure 7b, new Figure 8c, and new Figure 9h in the revised manuscript) and also refer to this quantification in the Results: page 12: lines 530-531; page 15: lines 641-643; page 16: lines 676-678; page 17: lines 762-765.

Reviewer #1

“While there is an association between neutrophil infiltration, Cit3H staining and enhanced pathology, the data do not establish direct causality between type I IFN signalling in neutrophils and NET formation. Alternative scenarios of drivers of NETosis may include increased neutrophil density in lung lesions (facilitated by type I IFN signalling) leading to elevated local ROS production or local hypoxia enhancing neutrophil elastase activity.”

Answer:

The idea that higher neutrophil density could promote NETosis is interesting. However, there are several experiments that argue strongly that the opposite is true. In vitro, NETosis efficiency decreases substantially (up to 70%) as cell confluence/density increases (Venizelos Papayannopoulos, unpublished). Moreover, fungal experiments indicate that yeast promote significantly higher neutrophil oxidation without inducing NETosis. By contrast hyphae promote NETosis with little detectable neutrophil oxidation due to ROS being pumped extracellularly. This suggests that the ROS threshold for activation of NET formation is low. Furthermore, in our model, in infected control mice there are numerous areas that contain a high local density of neutrophils, but NET formation is completely absent (new Figure 7a and data not shown). Additionally, we have now quantified the CitH3-positive stained area normalised to the MPO staining for each individual lung sections (new Figures 3f, new Figure 7b, new Figure 8c, and new Figure 9h in the revised manuscript)

These normalized quantifications demonstrate large increases in NET formation that far surpass the increase in neutrophil recruitment and suggest that the increase in NETosis is not merely a result of higher neutrophil infiltrates. Therefore, high neutrophil density does not seem to promote NETosis in our in vivo models of TB.

Reviewer #1

“The statistical analyses seem appropriate and the level of detail provided appears sufficient for others to independently reproduce these studies.”

Answer:

We thank the Reviewer for these positive comments.

Reviewer #1

“Other comments

1. Due to the small size of the figures, cells harbouring acid fast bacilli cannot be identified.”

Answer:

We thank the Reviewer for bringing this to our attention. We have now replaced the small size pictures by large pictures of Ziehl-Neelsen staining depicting cells harbouring acid fast bacilli in the *M. tuberculosis* infected lungs for the different experiments (new Figure 1g, new Figure 6e, and new Figure 9f in the revised manuscript).

Reviewer #1

“2. The MPO staining indicates a significantly different structure of the lung lesions in mice treated with anti-GM-CSF. Higher magnification analyses of the H&E stained sections should help identify whether this is representative of high density of intact neutrophils, neutrophils undergoing NETosis, or cellular debris in areas of necrosis.”

Answer:

We thank the Reviewer for pointing this out. We have now included a higher magnification picture of the H&E stained lung sections in the revised manuscript (Figure 1d, Figure 6b, and new Figure 9c). These higher magnifications illustrate the dense regions with NETosis (as confirmed by CitH3 staining) and cell debris (e.g., necrotic alveolar cells, alveolar macrophages and neutrophils) found in the lung lesions of infected C57Bl/6 WT mice treated with anti-GM-CSF mAbs (Figure 1d and Figure 6b) but absent in the lung lesions of infected mice treated with isotype control mAbs (Figure 1d and Figure 6b) and IFNAR-deficient mice (Figure 6b). In the TB-susceptible C3HeB/FeJ mice, we also observed these dense regions of NETosis and cell debris in the lung lesions of *M. tuberculosis* HN878 infected mice treated with isotype control mAbs but not in the lung lesions of infected mice treated with anti-IFNAR mAbs (new Figure 9c). In addition, we have also included the detailed H&E analysis for each of the individual lung sections from all independent experiments, scored by three board-certified veterinary pathologists blinded to the groups, in new Supplementary Data 1.

Reviewer #1

“3. Gene expression profiling analyses were done at one time point, 3 weeks post infection where clear differences in bacterial burden and pathology were observable and mice showed significant weight loss. Are there fundamental differences in blood and tissue gene expression, (neutrophil) infiltration, CFU, between control and anti-GM-CSF treatment earlier during infection? Do these correlate with the neutrophil/monocyte ratios in the blood?”

Answer:

We have chosen 3 weeks post-infection as the time-point to determine the impact of GM-CSF blockade on disease outcome following *M. tuberculosis* infection based on previous data from our laboratory. As shown in Figure 1b, *M. tuberculosis* infected mice treated

with anti-GM-CSF mAbs start showing signs of disease (e.g., noticeable weight loss) around 3 weeks post-infection. At 2 weeks post-infection we found no effect of GM-CSF blockade in lung bacterial replication, lung pathology or neutrophil infiltration into the site of infection (Philippa J. Stimpson, Evangelos Stavropoulos and Anne O'Garra, unpublished). In addition, we have previously established that time-points earlier than 3 weeks post *M. tuberculosis* infection were too early to reveal the whole blood transcriptional signature or even consistent lung transcriptional signatures in mouse models of TB as we have published (Moreira-Teixeira *et al.*, Nat Immunol 2020). Therefore, we chose 3 weeks post-infection for this study.

Reviewer #1

"4. Comparison with an *M. tuberculosis* isolate that induces lower type I IFN responses (e.g. CDC1551) might help further underpin the contributions of type I IFN to neutrophil infiltration in this model."

Answer:

There are numerous reports showing that such models do not show a phenotype of pathogenesis that is dependent on type I IFN signalling, unless type I IFN is induced by other means such as host genetic mutations in regulators of type I IFN (e.g., Tpl2, IL-1) or administration of adjuvants (e.g., poly(I)C) (Antonelli *et al.*, J Clin Invest 2010; McNab *et al.*, J Immunol 2013; Dorhoi *et al.*, Eur J Immunol 2014; Redford *et al.*, J Infect Dis 2014; Mayer-Barber *et al.*, Nature 2014; and reviewed in Moreira-Teixeira *et al.*, J Exp Med 2018). Moreover, we have recently published that resistant C57Bl/6 mice infected with *M. tuberculosis* laboratory strain H37Rv (known to induce lower type I IFN responses during infection) show very weak type I IFN signalling and weak neutrophilic signatures in the blood and lung and low number of neutrophils at the site of infection, compared to the TB-susceptible C3HeB/FeJ mice infected with *M. tuberculosis* clinical isolate HN878 (known to induce higher type I IFN responses during infection), which show strong type I IFN signalling and neutrophilic signatures (resembling human TB) and abundant neutrophil infiltration at the site of infection (Moreira-Teixeira *et al.*, Nat Immunol 2020). We now show in new Figure 9 and new Supplementary Data 1 in the revised manuscript, that type I IFN signalling also drives the neutrophil-induced disease exacerbation in the C3HeB/FeJ mouse model of increased susceptibility to TB, showing that this is a more generalised mechanism of TB pathogenesis over-and-above that of GM-CSF blockade.

Response to Reviewer #2:

We thank the Reviewer for reading the manuscript and providing detailed feedback. We provide below answers to all comments raised by the Reviewer and have made changes to the manuscript with provision of additional data as detailed below. We believe that the revised version has resulted in a much improved manuscript.

Reviewer #2:

“The O’Garra laboratory were the first to highlight the importance of blood gene expression biosignatures in human with expression of Type 1 interferons and genes associated with neutrophils. Here they have investigated the role of GM-CSF in mouse models of tuberculosis and in resected lung tissue from TB patients. Giving a blocking antibody to GM-CSF prevented any developmental issues in GM-CSF knock-out mice. Interestingly the authors did not find that blocking GM-CSF significantly affected T cell numbers or expression of IFN γ , previously hypothesised as a mechanism through which GM-CSF might act. Blocking GM-CSF increased the TB related transcriptional signature previously identified in humans.

Having demonstrated the involvement of Type I interferons, and genes associated with neutrophils in mice, biopsies from a group of patients with TB were studied, showing that NETs were present in the lung tissues. The TB patients studied were patients having lung resections at the end of their chemotherapy as they had refractory disease. To be able to study any human lung tissue, rather than just blood, in patients with tuberculosis is really valuable. However these were patients who were having lung resections as they had refractory disease. Did they have drug-resistant TB? Although the Abstract states that “NETs are also detected in necrotic lung lesions of TB patients showing a poor response to antibiotic therapy” the interpretation that this suggests “a broad role in TB pathogenesis” is rather exaggerated. A role for PMN associated NETs in TB has been proposed by others (see refs 62-66) but this is to my knowledge the first demonstration of NETs in TB lung tissue.”

Answer:

We thank the Reviewer for these positive comments and for recognizing the importance of our study as the first demonstration of NETs in lung tissue from patients with active TB. A role for PMN associated NETs in TB has been indeed previously proposed by other as we discussed in the “Discussion” section of our manuscript (page 21: lines 880-883 in the revised manuscript) but only using indirect NET-associated markers in the blood or other samples (Refs. 18, 63-67 in the revised manuscript). Our study is the first to provide direct demonstration of NETs in necrotic lung lesions of TB patients showing a poor response to antibiotic therapy (Figure 5 and Supplementary Figure 5a). Most of the patients in this study had multi and extensively drug-resistant TB but a few had drug-sensitive TB. The detailed information regarding drug-resistance is provided in Supplementary Data 6. Regarding the Reviewer’s concern that the interpretation that this suggests “a broad role in TB pathogenesis” is rather exaggerated, we agree and so have changed it to “supporting the role of NETs in a late stage of TB pathogenesis” (please see “Abstract”, page 2 in the revised manuscript) to ensure we do not over-state the message of the paper.

Reviewer #2:

“It may also be worth referencing a study published in 2017 in Front Immunol by Guimaraes-Costa et al, that showed that NETs treatment of IL-4 and GM-CSF treated monocytes affected their ability to deal with Leishmania parasites.”

Answer:

We thank the Reviewer for bringing this study to our attention, which we have now included in the “Discussion” section of our revised manuscript (Ref. 62 in the revised manuscript).

Reviewer #2:

“Minor comments

Line 95. Delete extra space and full stop.”

Answer:

We thank the Reviewer for bringing this to our attention. We have deleted the extra space and full stop.

Reviewer #2:

“Line 189-. Please clarify here that this analysis was performed in infected mice.”

Answer:

We have changed the text in our revised manuscript to: “genes differentially expressed between the lung samples from HN878 infected Ctrl Ab treated mice versus the lung samples from uninfected Ctrl Ab treated mice” (page 9: lines 375-377 in the revised manuscript). We apologise that this was not clear in the original manuscript.

Response to Reviewer #3:

We thank the Reviewer for reading the manuscript and providing detailed feedback. We provide below answers to all comments raised by the Reviewer and have made changes to the manuscript with provision of additional data as detailed below. We believe that the revised version has resulted in a much improved manuscript.

Reviewer #3:

“In this paper, Moreira-Teixeira et al seek to understand the contribution of GM-CSF signaling in the murine model of tuberculosis in a setting where alveolar macrophage development is not immediately disrupted. In this study, they utilized antibody-depletion to neutralize GM-CSF in vivo and compare this to animals treated with an isotype antibody. Nicely, the authors find similar results where loss of GM-CSF signaling enhances Mtb burden. The authors nicely demonstrate that this phenotype is independent of IFN γ R and IFN γ signaling. The authors next perform transcriptional analyses of the blood and the lung and compare these results to other publicly available datasets previously established in their groups. This drives the authors to hypothesize a role for neutrophils in the absence of GMCSF signaling. They then validate that treatment with a GMCSF antibody increases neutrophils as well as a decrease in alveolar macrophages. They nicely characterize neutrophil phenotype using the CitH3 antibody as a marker for netosis and see neutrophils have an altered phenotype during anti-GMCSF therapy. They see similar results in tissues obtained from humans with active TB undergoing surgery. They then further mine their data to implicate IFN signaling and show that loss of type I IFN receptor abrogates the altered bacterial burden observed in mice treated with the GMCSF antibody. They then show that deleting IFNAR in neutrophils results ameliorates the GMCSF phenotype.

In general, I highly enjoyed the paper and think the results are quite interesting. I think there are a few places where I think the authors could be a little more clear about their methodology.”

Answer:

We thank the Reviewer for these positive comments. We provide below answers to the specific comments related to the methodology raised by the Reviewer and have made changes to the manuscript to ensure that the methodology is clear (please see details below).

Reviewer #3:

“Minor: In referencing their experiments involving an isotype control, I think it would be helpful to say label the Ctrl as Isotype Ab for clarity.”

Answer:

We have now included “labelled as Ctrl Ab” in the Results, Methods and Figure Legends sections of the revised manuscript for clarity.

Reviewer #3:

“Minor: In Figure 1 e and 1f, is there an appropriate way to quantify the results statistically? Same for 6d.”

Answer:

We have now performed statistical analysis on the histopathological scoring of lung sections and provided the statistic results in the revised/new figures of our revised manuscript, as detailed below. The data now show pooled from five (revised Figure 1e and 1f), four (revised Figure 6c and 6d) or two (new Figure 9d and 9e) biological experiments and each dot represent an individual mouse (n= 2-4 mice/group/experiment). GraphPad Software was used for statistical analysis and Mann-Whitney test or Tukey’s multiple comparisons test were used to compare experimental groups, with $P < 0.05$ considered significant, as described in the revised figure legends. Detailed histopathological scorings of lung H&E staining for all experiments, including those representative experiments contributing to collective data, is provided in new Supplementary Data 1.

Reviewer #3:

“Minor: In Figure 1g, the figure doesn't adequately convey the message I think the authors are trying to make.”

Answer:

We thank the Reviewer for bringing this to our attention. As also noted by Reviewer #1, due to the small size of the pictures, cells harbouring acid fast bacilli could not be easily identified in the figures submitted with our original manuscript. We have now replaced the small size pictures by large pictures of the Ziehl-Neelsen staining depicting cells harbouring acid fast bacilli in the *M. tuberculosis* infected lungs for the different experiments (new Figure 1g, new Figure 6e, and new Figure 9f in the revised manuscript).

Reviewer #3:

“Minor — Lines 189 through 194 are a bit difficult to study methodologically. Are these lung homogenates? Specific cells from the lung? I had to go to the methods to understand this experiment, and think the authors could be a little more clear in the main text.”

Answer:

We thank the Reviewer for bringing this to our attention. We have now included the sentence “To determine the transcriptional response at the site of infection, RNA-seq data were obtained from whole lungs of the same mice used for the blood data from Fig. 2a.” in the “Results” section of our revised manuscript for clarity (page 9: lines 371-373 in the revised manuscript).

Reviewer #3:

“Minor: Was there anything else different about the neutrophils that the authors would like to comment on from their transcriptional analysis?”

Answer:

The transcriptional analysis of blood (new Supplementary Figure 3b) and lung (revised Figure 3b) samples revealed that the expression of genes associated with neutrophil recruitment but also activation, such as *S100a6*, *S100a8*, *S100a9*, *Mmp8*, *Mmp9*, *Cd177* and *Lcn2*, was increased in the blood and lungs from infected mice treated with anti-GM-CSF mAbs compared to infected Ctrl mice (new Supplementary Figure 3b, revised Figure 3b and Supplementary Data 3 and 5). Thus, suggesting that GM-CSF blockade during *M. tuberculosis* infection affects not only the recruitment but also the activation status of neutrophils in circulation and at the site of infection. We have now discussed these observations in more detail in the “Results” (page 11: lines 484-497 in the revised manuscript) and “Discussion” (page 20: lines 860-870 in the revised manuscript) sections of our revised manuscript.

Reviewer #3:

“Major: Is there any quantification that the authors can do to characterize their histopathology results? It's a little hard to think about changes in neutrophil abundance and state simultaneously.”

Answer:

We thank the Reviewer for bringing that to our attention. We have now quantified and provided as additional data in the revised version of our manuscript the percentage of CitH3-positive stained area within MPO-positive stained area for individual lung sections of each independent experiment (new Figures 3f, new Figure 7b, new Figure 8c, and new Figure 9h in the revised manuscript). These normalized quantifications demonstrate large increases in NET formation that far surpass the increase in neutrophil recruitment upon GM-CSF blockade during *M. tuberculosis* infection or in TB-susceptible C3HeB/FeJ mice, suggesting that the observed NETosis is not merely a result of higher neutrophil abundance. In addition, detailed histopathological scorings of lung H&E staining, including neutrophil abundance, for each independent experiment are provided in new Supplementary Data 1 in the revised manuscript.

Reviewer #3:

“Major: Are there any other tissues/samples that the authors have access to that show the robustness of their antibody?”

Answer:

We presume that the Reviewer is referring to our anti-GM-CSF mAb (clone MP1-22E9 from DNAX [now Merck, USA]). We have reproduced our experiments using anti-GM-CSF mAb purchased from BioXCell (clone MP1-22E9) with similar results. Moreover, we have performed the same experiment as detailed in Figure 1a using GM-CSF-deficient mice and

we did not observe any differences in weight loss, lung bacterial growth and lung pathology between *M. tuberculosis* infected GM-CSF-deficient mice treated with anti-GM-CSF or isotype ctrl mAbs (unpublished), confirming the specificity of our anti-GM-CSF mAb.

Reviewer #3:

“Major: The authors make the jump from a transcriptional signature about IFN signaling in the lung without any protein validation for evidence of enhanced production of the cytokine during GMCSF antibody treatment. Did the authors see any evidence of this that they can comment on?”

Answer:

We appreciate the Reviewer’s concern. However, it is known throughout the field that type I IFNs are very tightly regulated and that protein is often hard to detect. Therefore, through the years our group and many others have relied on the expression of type I IFN-inducible genes as markers of type I IFNs. Moreover, herein we block IFNAR activity by using IFNAR-deficient mice (Figure 6 and Figure 7 in the revised manuscript) or anti-IFNAR mabs (new Figure 9 in the revised manuscript) with a clear effect on the phenotype observed, therefore indicating that there must have been protein produced following *M. tuberculosis* infection in our models.

Reviewer #3:

“Major: While I recognize the power of their transcriptional signatures that they wanted to compare to, I think the authors missed an opportunity to speak more broadly about the transcriptomic data before transitioning to the analysis of the signatures. For example, I think they could have brought a little bit more of Supplemental Figure 2 into the main text to talk about the results more broadly.”

Answer:

Also prompted by Reviewer #1, we have now extended the analysis of our blood RNA-seq data and described these transcriptomic data more broadly together with the analysis of the signatures (page 8: lines 325-332 and 344-348 in the revised manuscript). Using cellular deconvolution analysis of our blood RNA-Seq transcriptomic data (new Supplementary Figure 3a in the revised manuscript), a technique which has previously been shown to mirror changes in cellular composition detected by flow cytometry (Singhania *et al.*, Nat Commun 2018), we show here that a similar immune cell composition was observed in uninfected mice treated with anti-GM-CSF (aGM-CSF) or isotype control (ctrl) mAbs for 3 weeks. During *M. tuberculosis* infection we observed an increase in the abundance of neutrophils in 3 out of 5 blood samples but this was accompanied by an increase in the overall abundance of all other myeloid (monocytes and macrophages) cells in all blood samples obtained from infected mice treated with aGM-CSF compared to ctrl mAbs (new Supplementary Figure 3a). Transcriptomic analysis of the blood samples, showed that expression of genes not only associated with neutrophils (new Supplementary Figure 3b) but also monocytes (new Supplementary Figure 3c) and macrophages (new Supplementary Figure 3d) were increased upon GM-CSF blockade in

M. tuberculosis infected mice. Together with the increased over-abundance of myeloid and granulocyte-associated modules in the blood of those mice (Figure 2a), our new data show that all of these populations increased in the blood during *M. tuberculosis* infection. in the absence of GM-CSF. On the other hand, cellular deconvolution analyses show a decrease in the percentage of B and CD4+ T cell fractions in the blood from infected aGM-CSF-treated compared to infected Ctrl mice (new Supplementary Figure 3a in the revised manuscript), in keeping with the lower abundance of the B cell and T cell modules (Figure 2a) and also reflected by decreased expression of key T and B cell-specific genes (Supplementary Data 2 and Supplementary Figure 2).

REVIEWERS' COMMENTS

Reviewer #1 (Remarks to the Author):

The new data, especially expansion beyond the GM-CSF depletion, as well as the clarifications have further enhanced the message and clarity of this manuscript. All of my queries and concerns have been addressed. Congratulations to the authors on this excellent study.

Reviewer #3 (Remarks to the Author):

The authors have adequately addressed my concerns.

REVIEWERS' COMMENTS

Reviewer #1 (Remarks to the Author):

The new data, especially expansion beyond the GM-CSF depletion, as well as the clarifications have further enhanced the message and clarity of this manuscript. All of my queries and concerns have been addressed. Congratulations to the authors on this excellent study.

Answer:

We thank the Reviewer for acknowledging that we have adequately addressed all previous queries and concerns, and for recognising that the new data and clarifications have further enhanced the message and clarity of our manuscript.

Reviewer #3 (Remarks to the Author):

The authors have adequately addressed my concerns.

Answer:

We thank the Reviewer for acknowledging that we have adequately addressed all previous concerns.